# Trained immunity modulates inflammation-induced fibrosis

Mohamed Jeljeli[1,2], Luiza Gama Coelho Riccio [1,3], Ludivine Doridot[1], Charlotte Chêne[1], Carole Nicco [1], Sandrine Chouzenoux[1], Quentin Deletang[1], Yannick Allanore [1,4], Niloufar Kavian[1,2,5] & Frédéric Batteux[1,2]*

Chronic inflammation and fibrosis can result from inappropriately activated immune responses that are mediated by macrophages. Macrophages can acquire memory-like characteristics in response to antigen exposure. Here, we show the effect of BCG or low-dose LPS stimulation on macrophage phenotype, cytokine production, chromatin and metabolic modifications. Low-dose LPS training alleviates fibrosis and inflammation in a mouse model of systemic sclerosis (SSc), whereas BCG-training exacerbates disease in this model. Adoptive transfer of low-dose LPS-trained or BCG-trained macrophages also has beneficial or harmful effects, respectively. Furthermore, coculture with low-dose LPS trained macrophages reduces the fibro-inflammatory profile of fibroblasts from mice and patients with SSc, indicating that trained immunity might be a phenomenon that can be targeted to treat SSc and other autoimmune and inflammatory fibrotic disorders.

[1] Institut Cochin, INSERM U1016, Université Paris Descartes, Sorbonne Paris Cité, Paris, France. [2] Assistance Publique–Hôpitaux de Paris (AP–HP), Hôpital Universitaire Paris Centre (HUPC), Centre Hospitalier Universitaire (CHU) Cochin, Service d'immunologie biologique (Professeur Batteux), Paris, France. [3] Disciplina de Ginecologia, Departamento de Obstetrícia e Ginecologia, Faculdade de Medicina FMUSP, Universidade de São Paulo, São Paulo BR, Brazil. [4] Assistance Publique–Hôpitaux de Paris (AP–HP), Hôpital Universitaire Paris Centre (HUPC), Centre Hospitalier Universitaire (CHU) Cochin, Service de rhumatologie, Paris, France. [5] Hong Kong University-Pasteur Research Pole, School of Public Health, The University of Hong Kong, Hong Kong, China. *email: frederic.batteux@aphp.fr

nflammation is a physiological defense mechanism against injurious stimuli of infectious, toxic, or immune origin. Timely inflammation in adequate intensity is essential to eliminate harmful stimuli. Yet, if the trigger remains, an inappropriate chronic inflammatory response occurs, resulting in progressive fibrosis with an excessive accumulation of extracellular matrix (ECM) components[1] and leading to disrupted tissue function, organ failure, and mortality. Innate immune responses play key roles in initiation of healing processes especially macrophages that orchestrate outcomes of inflammation and fibrosis[2]. Early on during the pathogenesis of fibrosis, macrophages migrate to the inflammation site and establish a complex communication system with ECM-producing myofibroblasts and endothelial cells[3]. Several profibrotic mediators including transforming growth factor-β (TGF-β) and various cytokines/chemokines are produced by macrophages to favor myofibroblasts differentiation, ECM production, and inflammatory cells migration[4]. Thus, macrophages appear to be implicated into all stages of the fibrotic process and exhibit transitions in phenotype and function as tissue repair progresses[5]. Recently, the concept of "trained immunity" has emerged and shed a new light in macrophages function. In contrast to the classical concept that assigned immune memory only to the adaptive immune system, Netea et al. have demonstrated that macrophages and other innate immune cells can also acquire a type of immune memory as their secondary response to an antigenic challenge with microorganisms component differ in quality and intensity compared to the first encounter with another or the same pathogens[6–8]. Clinical and experimental observations support this idea. After a first challenge with the BCG vaccine, macrophages can exhibit an enhanced function with heightened inflammatory response when exposed to the same pathogen or to an unrelated second stimulation[9–11]. Contrarily, macrophages can also be trained to become unresponsive and immunotolerant by treatment with endotoxin[12–14]. Endotoxin tolerance has been observed in patients with septicemia for decades and has shown to cause severe secondary infection due to the unresponsiveness of innate immune cells upon a second exposure to the same bacteria or to other types of pathogens[15]. The molecular mechanism of innate immune memory involves metabolic and epigenetic reprogramming with chromatin modifications (H3K4me1, H3K27Ac, H3K9me2, and H3K4me3)[16–19] on promoters of genes encoding cytokines, signaling proteins, and surface markers. While the involvement of trained innate immunity was suggested in inflammatory and autoimmune diseases[20], a demonstration is still lacking. Moreover, using trained immunity to modulate the development of such diseases was never attempted. Based on the experimental evidence that macrophages can be trained, with long-lasting effects on one hand, and the well-established role of macrophages in the pathogenesis of numerous fibro-inflammatory chronic affections on the other[21–24], we tested the effects of BCG- and LPS-(Endotoxin)-trained macrophages in a mice model of systemic fibrosis with an autoimmune feature. This way, we demonstrate that both in vivo training and cellular therapy with trained macrophages can slow down or speed up inflammation and fibrosis, as well as reduce or increase the production of auto-antibodies in mice, the specific effect being dependent on the type of training. Furthermore, these effects are relevant to humans as fibroblasts from patients with systemic sclerosis, a chronic autoimmune inflammatory disease characterized by extensive fibrosis of the skin and visceral organs showed a similar dampening of the fibrotic phenotype following a co-culture with trained human macrophages. Our results do not only provide new dynamics aspects on the role of trained macrophages in the pathogenesis of fibro-inflammatory disorders, but also open the way to design original therapeutic strategies based on trained macrophages to counteract fibro-inflammatory and autoimmune diseases.

## Results

### In vivo immune training affects macrophages response.
To better characterize the phenotypic changes upon in vivo immune training with LPS$^{low}$ or BCG (Fig. 1a), we evaluated several cell surface molecules that are potentially expressed by macrophages in various states of their activation. Compared to the Controls (PBS-mice), splenic macrophages from BCG-mice displayed an enhanced expression of CCR2, CXCR4 (Fig. 1b) TLR2, TLR4 (Fig. 1c), Ly6C, CD43 (Fig. 1d), CD80 (Fig. 1e), CD206 (Fig. 1f), CD40, CD68, and a trend toward an increase in MHC-class II expression (Supplementary Fig. 2). No changes were observed in the expression of DC-SIGN, iCOS ligand, and CD93 in BCG-mice compared to Controls (Fig. 1e, f and Supplementary Fig. 2). In contrast, LPS$^{low}$-training induced a decrease of Ly6C (Fig. 1d), CD68, and MHC-II expression (Supplementary Fig. 2). Interestingly, LPS$^{low}$-mice showed an enhancement of DC-SIGN and iCOS ligand compared to BCG-trained mice and controls (Fig. 1e, f). Importantly, macrophages from LPS$^{low}$-trained mice showed a similar increase in CD43 and CD206 compared to controls as observed in BCG-mice (Fig. 1d, f). These phenotypes are challenging the dualistic concept of classic (M1) or alternative (M2) polarization, as BCG-training upregulated the M1 markers CD43, CD68, CD80, and Ly6C as well as the M2 marker CD206 showing a singular activation of trained macrophages. Dynamic changes in cytokine production after in vivo immune training were assessed by stimulating splenic macrophages of trained-mice with an inflammatory dose of LPS referred as LPS$^{high}$ (Fig. 1h). Ex vivo PBS-macrophages stimulation led to increased levels of IL-1β, TNF, IL-6, and IL-10 compared to basal cytokine production in unstimulated PBS-macrophages. BCG-macrophages showed an enhanced pro-inflammatory cytokine production after LPS$^{high}$ stimulation compared to stimulated PBS-macrophages (IL-1β $p = 0.04$; TNF $p = 0.006$; IL-6 $p < 0.0001$, unpaired $t$-test). By contrast, stimulated LPS$^{low}$-macrophages showed a significant decrease in their production of IL-1β, TNF, and IL-6 but an important increase of IL-10 production. Thus, in vivo training of macrophages markedly modified their cytokines production upon inflammatory challenge with an increased responsiveness of BCG-macrophages on one hand and a dramatic decrease in the secretion of pro-inflammatory cytokines with an upregulation of IL-10 secretion by LPS$^{low}$-macrophages on the other hand. As the reprogramming of cellular metabolism is a hallmark of trained immunity[9,15,18], we evaluated the lactate production that reflects glycolysis activity in the supernatant of trained splenic macrophages stimulated with LPS$^{high}$. Lactate release from BCG-macrophages was significantly increased compared to the PBS-macrophages while LPS$^{low}$-training reduced its production (Fig. 1g).

Epigenetically, BCG-training was associated with an increased H3K4 trimethylation on the IL-6 and TNF promoter when compared PBS-training. By contrast, macrophages from LPS$^{low}$-mice displayed a significant increase of histone H3K4 trimethylation at IL-10 loci vs BCG- and PBS-mice (Fig. 1h). These chromatin changes are consistent with the differences observed with the cytokine secretion upon stimulation of trained macrophages.

### Opposite effect of LPS$^{low}$ or BCG training on SSc-mice.
To assess how immune training could influence the course of inflammation and the development of the skin and visceral fibrosis, mice were trained with BCG, LPS$^{low}$, or PBS and subjected to HOCl injection to induce systemic fibrosis. HOCl-injections generated Reactive Oxygen Species locally and constituted a second inflammatory trigger after mice training. One group of PBS-mice remained untreated with HOCl (Controls).

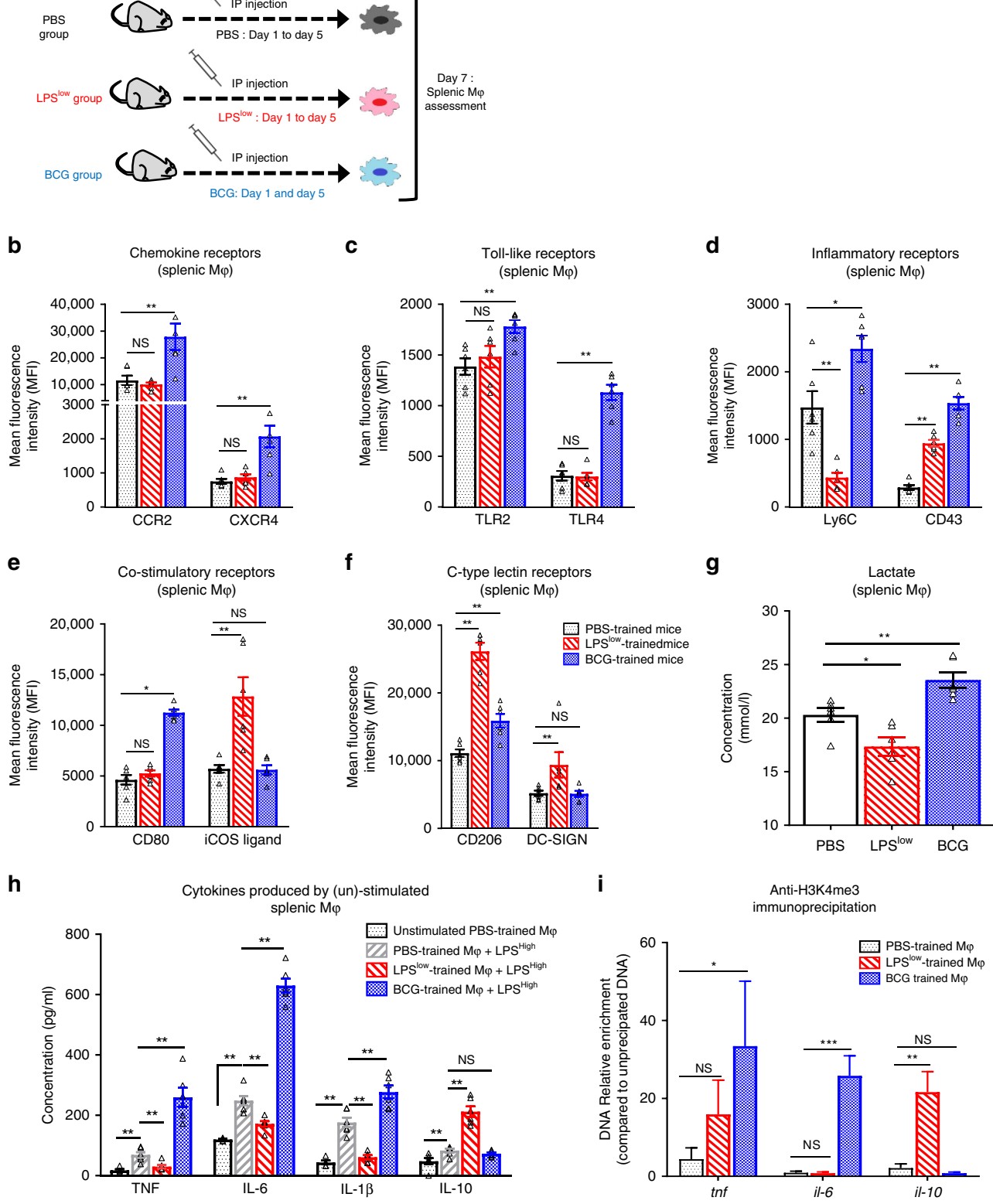

HOCl-treated mice were referred as HOCl-PBS, HOCl-BCG, and HOCl-LPS$^{low}$ (Fig. 2a).

At day 22 of the experiment, corresponding to the peak of the acute inflammatory phase, HOCl injection induced an upregulation of CD206 on dermal macrophages from all groups compared to the Controls. Dermal macrophages from HOCl-PBS mice displayed elevated expression of CCR2 and Ly6-C compared to

controls and even markedly in BCG-trained mice. Expression of CXCR4 was highly upregulated on skin macrophages from HOCl-BCG mice compared to HOCl-PBS. By contrast, macrophages from HOCl-LPS$^{low}$ mice showed a significant decrease in CCR2 and Ly6C expression compared to HOCl-PBS, but an elevated iCOS ligand and DC-SIGN expression compared to all groups (Fig. 2b). Given the important role of the CD4 + T and B

**Fig. 1 Effects of in vivo immune training on macrophage responses. a** Experimental setup of the in vivo training model used ($n = 6$ mice per group).
**b–f** Flow cytometric analysis of splenic macrophage phenotype at day 7 after training. Spleen cell suspensions were prepared after hypotonic lysis of erythrocytes in potassium acetate solution and three washes in complete RPMI medium. Spleen cells were counted and Fc receptors were blocked with 10 μg/mL anti-CD16/CD32 antibody. Quantitative analysis of surface markers expression was performed on cells gated on CD11b$^+$ F4/80$^+$ representing the macrophage population. Data represent the mean fluorescence index (MFI) and SEM obtained from $n = 6$ biologically independent mice. **g** Lactate production of trained splenic macrophages stimulated with LPS (100 ng/ml). **h** Adherent splenic-derived macrophages were stimulated with 100 ng/ml LPS. Cytokine production (IL-1β, TNF, IL-6, and IL-10) was assessed by ELISA in culture supernatant after 24 h of incubation. Each box represents mean ± SEM of triplicate obtained with cell culture from $n = 6$ biologically independent mice. **i** ChIP analysis of H3K4me3 enrichment at the promoter of *Tnf, Il6*, and *Il10* in adherent splenic macrophages isolated from the different trained groups. Data represent the mean of relative expression and SEM (normalized to the input un-precipitated DNA). Data represent the mean of relative expression and SEM versus the input (un-precipitated DNA). The ANOVA test with Bonferroni correction was used to detect significant differences between the groups. Mφ: Macrophages. NS: Non significant; *$p ≤ 0.05$; **$p ≤ 0.01$; ***$p ≤ 0.001$. Source data are provided as a Source Data file.

cells in the pathogenesis of SSc, we analyzed their number and activation status in the spleen (Fig. 2c and Supplementary Fig. 3a). Disease induction increased the number of splenic CD4 + T and B cells in all groups, though to a different level according to the in vivo training. Activation of TCD4 + and B cells was significantly increased in HOCl-BCG group compared to HOCl-PBS. LPS$^{low}$-training was associated with a reduced expression of CD40 on splenic B cells and a decrease of both absolute count and CD69 expression on CD4 + T cells.

We then measured cytokine production (Fig. 2d and supplementary Fig. 3b) following an inflammatory stimulus both at the systemic (splenic) and local (dermal-infiltrated immune cells) levels. HOCl-PBS mice had a significant increase in splenic production of IL-6, IL-4, and IL-13 but no difference of IL-1β and TNF compared to controls. By contrast, BCG-mice displayed an enhanced production of IL-6, IL-1β, TNF, and IL-4 by spleen cells compared to HOCl-PBS group while HOCl-LPS$^{low}$ mice showed a significant decrease of IL-4, IL-13, and IL-6 secretion, and a slight decrease of IL-1β production. At the dermal level, BCG-training enhanced the production of CCL-2 compared to HOCl-PBS group while LPS$^{low}$-mice showed a significant decreased production of IL-17 and CCL-2 but a markedly enhanced IL-10 compared to HOCl-PBS group (Fig. 2d). Altogether, these data highlight the determinant effect that differential in vivo training may exert on immune cells in terms of activation, frequency, and cytokine production in the context of an inflammatory disease.

As the inflammatory phase is followed by the establishment of fibrosis, the remaining groups of mice were euthanized and analyzed at day 42 of the experiment. Weekly measurement of the skinfold showed that BCG-training induced an acceleration of skin thickening compared to HOCl-PBS mice (1/slope = 24.98 vs 35.85, $p = 0.01$, skinfold: 2.16 ± 0.16 mm vs 1.67 ± 0.16 mm, $p < 0.0001$, unpaired $t$-test, Fig. 3a). Conversely, HOCl-LPS$^{low}$ mice had a reduction of fibrosis compared with HOCl-PBS mice (1/slope = 67.93 vs 35.85, $p = 0.02$; skinfold 1.15 ± 0.05 mm vs 1.67 ± 0.16 mm, $p < 0.0001$, unpaired $t$-test). HOCl-BCG mice showed a significant increase in collagen content in the skin and lung as well as increased dermal and pulmonary mRNA expression of *Co1a1*, *α-sma* and *Tgfb1* in skin while LPS$^{low}$-training induced a significant decrease of hydroxyproline content, mRNA expression of *Col1a1* and *Tgfb1* in the skin and lung and a reduction of the dermal mRNA expression of *Il-13* and *α-Sma* compared to the HOCl-PBS group (Fig. 3b–e, h).

Beyond markers of fibrosis, the seric levels of AOPP and anti-DNA-topoisomerase I autoantibodies that are systemic markers of the diffuse form of the disease, tended to be elevated in HOCl-BCG mice (Fig. 3f, g) in contrast to HOCl-LPS$^{low}$ mice (Fig. 3f, g). Thus, LPS$^{low}$-training prior to the disease induction HOCl significantly reduced the establishment of fibrosis as reflected by a lower production of skin and lung collagen while BCG training worsened the clinical and biological features of the disease. These

data support the modulatory effects of in vivo immune training on inflammation and fibrosis through the dynamic modifications of macrophages' phenotype, T cell activation, and cytokine production.

**Dual interaction of trained macrophages and fibroblasts**. To further decipher the nature of cross-communication between trained macrophages and fibroblasts in the fibro-inflammatory process, we established a co-culture model using trained BMDM and either resting fibroblasts or activated HOCl-fibroblasts obtained from the skin of PBS or HOCl-treated mice, respectively (Fig. 4a). HOCl-fibroblasts induced an enhancement of CCR2, CXCR4, Ly6C, CD206, CD80, and CD43 expression on BCG-macrophages but failed to increase the expression of these markers on LPS$^{low}$-macrophages except the CD206 marker (Fig. 4b). Of note, resting fibroblasts did not affect the surface expression of the assessed markers whatever the training of macrophages is (Supplementary Fig. 4).

We then analyzed the effects that trained macrophages may exert on the HOCl-fibroblasts phenotype. The surface adhesion marker CD44 and α-sma (hallmark of fibroblast-to-myofibroblast differentiation) mRNA expression were enhanced on fibroblasts from HOCl-mice compared to fibroblasts from PBS-mice. The co-culture of HOCl-fibroblasts with PBS-macrophages did not influence their expression. Interestingly, BCG-macrophages enhanced CD44 surface expression and α-sma mRNA synthesis in HOCl-fibroblasts while LPS$^{low}$-macrophages downregulated their expression (Fig. 4c, d).

We then focused on the secretion of soluble immunological factors. Basal cytokines production of trained macrophages cultured alone and with non-activated fibroblasts is shown in Supplementary Figs. 4b and 5a. The level of IL-1β, TNF, IL-6, TGF-β, and CCR2 was significantly increased in the supernatant of BCG-macrophages in co-culture with HOCl-fibroblasts compared to PBS-macrophages with HOCl-fibroblasts (Fig. 4e). By contrast, the production of IL-1β, TNF, IL-6, and CCR2 in the supernatant of LPS$^{low}$-macrophages co-cultured with HOCl-fibroblasts was significantly decreased. These data may reflect the complex cross-talk between macrophages and fibroblasts in the process of fibrogenesis where myofibroblasts act as an inflammation trigger to the trained macrophages and shape their activation status and cytokine production. In return, trained macrophages may modulate the phenotype of myofibroblasts and behave like a potentiator or antagonist, according to the nature of the immune training.

**Adoptive transfer of LPS$^{low}$ macrophages alleviates HOCl-induced SSc**. To ascertain the role of macrophages training in the course of inflammatory-induced fibrosis and to translate our results into a pre-clinical cellular therapy protocol, we evaluated

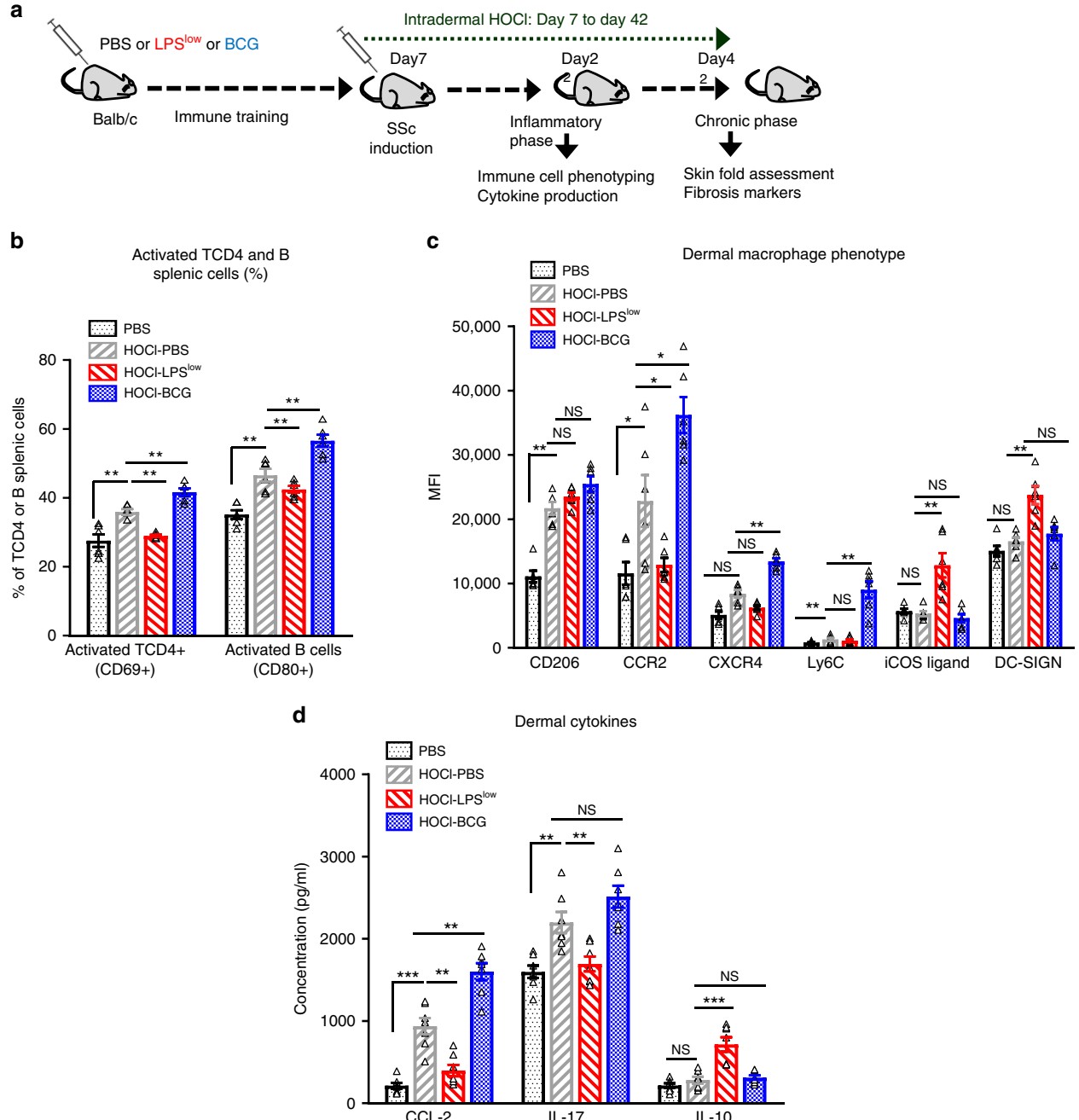

**Fig. 2 Trained immunity modulates inflammation in HOCl-induced SSc. a** Schematic representation of the experimental induction of fibrosis in trained mice. Mice ($n = 12$ per group) were trained either by LPS[low], BCG, or PBS and then injected intradermally from day 8 to day 42 with HOCl. The control group (PBS mice) was injected daily with sterile PBS for 6 weeks. At day 22 of the experiment (Acute inflammatory phase), 7 animals per group were euthanized to perform immune cell phenotyping and cytokine production assessment. The remaining animals ($n = 5$ per group) were euthanized at day 42 (Chronic fibrotic phase) to perform fibrosis markers assessment. **b** Flow cytometric characterization of splenic B cells and CD4[+] T cells at day 22. Data represent the absolute count for T CD4[+] and B cells and percentage of activated B cells (Expression of CD80) and CD4[+] T cells (Expression of CD69) with SEM from $n = 5$ biologically independent samples. **c** Flow cytometric analysis of dermal macrophage phenotype at day 22. Dermal macrophages were gated on CD11b and F4/80 positive cells among CD45 positive. Data represent the mean fluorescence index (MFI) and SEM from $n = 7$ biologically independent mice. **d** Cytokine production (pg/ml) in dermal culture supernatant (ELISA assessment). Each box represents mean ± SEM of triplicate obtained with cell culture from $n = 7$ biologically independent mice. Statistics are shown between PBS and HOCl-PBS groups first and then treated groups (LPS[low] and BCG) versus HOCl-PBS group. The ANOVA test with Bonferroni correction was used to detect significant differences between the groups. NS: Non significant; *$p \leq 0.05$; **$p \leq 0.01$; ***$p \leq 0.001$. Source data are provided as a Source Data file.

the therapeutic potential of a cellular therapy based on a weekly in vitro trained BMDM injection on HOCl-mice (Fig. 5a). Peritoneal injections of PBS-trained macrophages did not impact the skin thickening in HOCl-mice (considered as controls in this experiment). Injection of LPS[low]-macrophages induced a slower

progression of skin thickening compared to the controls while treatment with BCG- macrophages accelerated it (Fig. 5b). Collagen content in skin was elevated in mice injected with BCG-macrophages and reduced in animals treated with LPS[low]-macrophages when compared to HOCl-mice (Fig. 5c). A similar trend

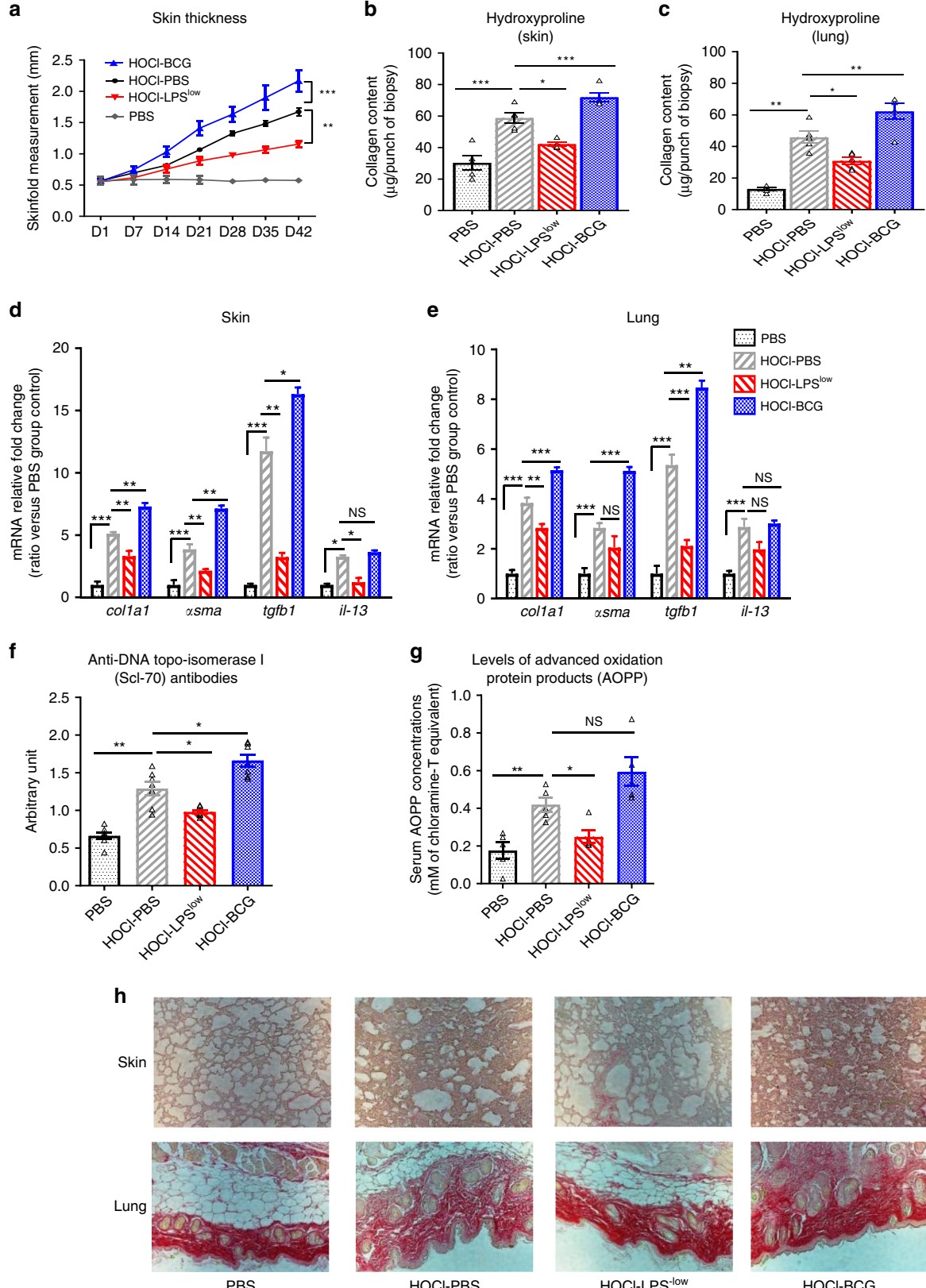

of variation was observed in lungs, although not significant (Supplementary Fig. 6a).

Fibrosis markers correlated with the clinical findings as we observed a significant increase of α-sma, Col1a1, and Tgfb1 mRNA expression in the skin of mice treated with BCG-macrophages, while these markers were decreased in mice treated with LPS$^{low}$-macrophages compared to HOCl-mice (Fig. 5d). In the lung, treatment with BCG-treated macrophage induced an overexpression of Tgfb1 and a trend toward an increase of α-sma (Supplementary Fig. 6b), while administration of LPS$^{low}$-macrophages downregulated these two markers. Seric circulating anti-Scl70 antibodies and AOPP levels showed a similar profile

**Fig. 3 Trained immunity modulates fibrosis in HOCl-induced SSc. a** Evolution of skin fold thickness in millimeters from day 1 to day 42, measured weekly. **b** Collagen type I content in skin (Hydroxyproline dosage, mg/punch biopsy). **c** Collagen type I content in lung (Hydroxyproline dosage, mg/lobe biopsy). Each box represents mean ± SEM from $n = 5$ biologically independent mice. **d**, **e** Col1a1, α-sma, Tfgb1, and Il13 mRNA levels in skin (**d**) and lung (**e**). Results were expressed as fold increase versus the PBS control group, derived from $n = 5$ biologically independent mice (Unpaired T-test). **f** Concentrations of anti-DNA topo-isomerase I (Scl70) antibodies in the sera from mice (Arbitrary Unit). Each box represents mean ± SEM from $n = 5$ biologically independent mice. **g** Concentrations of advanced oxidation protein products (AOPP) in the sera from mice (mM of chloramine T equivalent). Each box represents mean ± SEM from $n = 5$ biologically independent mice. **h** Representative Sirius red–dyed skin and lung sections of 6 μm, showing enhanced fibrosis in HOCl-BCG mice. Photographs were taken with a Nikon Eclipse 80i microscope. Original magnification ×20. The ANOVA test with Bonferroni correction was used to detect significant differences between the groups (unless stated). NS: Non significant; *$p \leq 0.05$; **$p \leq 0.01$; ***$p \leq 0.001$. Source data are provided as a Source Data file.

with a significant increase in HOCl-mice injected with BCG-macrophages and a decrease in mice injected with LPS[low]-macrophages compared to controls (Fig. 5e, f). Then, we analyzed the phenotype of skin-derived macrophages and level of dermal and seric cytokine production. Expression of CCR2, Ly6C, CD206, and iCOS Ligand was similar in untreated HOCl-mice and HOCl-mice treated with PBS-macrophages. Injection of BCG-macrophages increased the expression of both CCR2 and Ly6C receptors when compared to HOCl-mice. On the contrary, LPS[low]-macrophages significantly decreased the expression of CCR2 and to a lesser extent in Ly6C but upregulated iCOS ligand expression on skin-derived macrophages (Fig. 5g). LPS[low]-macrophages injection significantly increased the seric concentration of IL-10 while it reduced the capacity of IL-6 production at the dermal level. By contrast, treatment of HOCl-mice with BCG- macrophages increased IL-6 production by dermal cells compared to HOCl-mice and also increased the level of IL-10 in sera, although to a lesser extent than LPS[low]-macrophages, (Fig. 5h, i). Altogether, these results provide strong evidence that treatment with trained-macrophages is able to modulate the course of inflammation and fibrosis in this experimental model of SSc, as reflected by a significant clinical impact and immunological effects that are consistent with our previous in vivo observations confirming the prominent role of macrophages in this process.

**Trained macrophages interact with human SSc-fibroblasts**. The impact of macrophages training on fibrosis was then evaluated in a co-culture model using diseased human SSc-fibroblasts with trained macrophages from healthy controls (HC). Levels of IL-6, CCL-2, and TNF were considerably increased in the supernatant of SSc-fibroblast cultured with BCG-macrophages compared to SSc-fibroblasts cultured alone or with PBS-macrophages. LPS[low]-macrophages reduced IL-6 and CCL-2 levels in the supernatant but did not influence the production of TNF. As observed in the mice experiments, the level of IL-10 was significantly increased when LPS[low]-macrophages were co-cultured with diseased SSc-fibroblasts compared to BCG- and PBS-macrophages (Fig. 6a). Regarding fibrosis markers, mRNA expression of α-sma, Col1a1, Cd44, Icam1, Ctgf, and Serpin1 was significantly increased in SSc-fibroblast compared to fibroblast from healthy controls (HC). Co-culture of PBS-macrophages with SSc-fibroblasts did not influence the expression of these genes compared to SSc-fibroblasts cultured alone. However, when cultured with SSc-fibroblasts, BCG- macrophages significantly increased the expression of these markers (except a tendency toward an increase for Ctgf), while LPS[low]-macrophages notably reduced their expression (Fig. 6b–g). Basal cytokine production in unstimulated trained macrophages cultured alone was similar (Supplementary Fig. 7a). Also, fibroblasts from HC co-cultured with trained macrophages did not show any differences in the cytokine production levels (Supplementary Fig. 7b) or fibrosis markers, except for Icam-1 that was upregulated in the BCG-macrophages-fibroblasts co-

culture but to a lesser extent than that observed in the co-culture with SSc-fibroblasts (Supplementary Fig. 8). These results were confirmed by a separate experiment conducted with another SSc-fibroblasts lineage from a second SSc patient (Supplementary Fig. 9). These data suggest that LPS[low]-macrophages can dampen in vitro the fibro-inflammatory phenotype of diseased human SSc-fibroblasts paving the way for the use of trained macrophage-based cellular therapy for this condition in humans.

## Discussion

The concept of trained immunity offers interesting perspectives in vaccine development and treatment strategies in infectious diseases. However, in autoimmune and fibro-inflammatory diseases, its effect is unexplored. Here, we show that trained immunity can have different functions either in accelerating or attenuating inflammatory processes according to the nature of the training, as innate cells such as macrophages are usually at the first line of the pathogenesis in these diseases[25]. We first showed an opposite activation status and a singular phenotype, cytokine production and epigenetic profile of trained macrophages dependent on stimuli used for training (BCG or LPS[low]). Data on the phenotype of BCG or LPS[low]-trained macrophages are scarce. A previous study showed that enhanced non-specific responses to infections were accompanied by heightened expression of CD14, TLR4, and CD206 receptors on macrophages upon BCG vaccination in human[11] and increased mRNA expression of Tlr2, Tlr4, Cd163, and Cd206 upon β-Glucan challenge[8]. In our in vivo training model, we extended the panel of markers and showed an enhancement of the expression of various pattern recognition receptors (TLR4, CD206, and CD14) chemokine receptors (CCR2 and CXCR4) and co-stimulatory and/or signaling molecules (CD43, CD14, CD40, CD80, CD68, and Ly6C). These receptors are known to favor inflammation, T cell stimulation, immune cell trafficking, angiogenesis and wound healing[26–28]. By contrast, LPS[low]-macrophages had lower expression of costimulatory receptors, but upregulated iCOS-ligand that is involved in skin wound healing[29] and of DC-SIGN whose expression on macrophages has recently been implicated in the induction of post-transplantation tolerance[30]. These changes in phenotype expression of trained macrophages do not fit the classic M1 (classical) and M2 (alternative) macrophage polarization concept[31] reflecting more specifically the complex in vivo environment in which macrophages can acquire a mixed activation status upon antigen-mediated training. Altogether, these results offer a more extensive description of the phenotypic profiles that can be used to better characterize and track trained macrophages. Functional evaluation of the in vivo BCG-macrophages showed a potent capacity to release the pro-inflammatory cytokines IL-6, TNF, and IL-1β in line with previous observations[10], whereas LPS[low]-macrophages showed an impairment of pro-inflammatory cytokine production but an enhanced IL-10 release capacity. Clinical observations demonstrated that chronic exposure to LPS, such as during the late phase of sepsis can be associated with immunoparalysis and

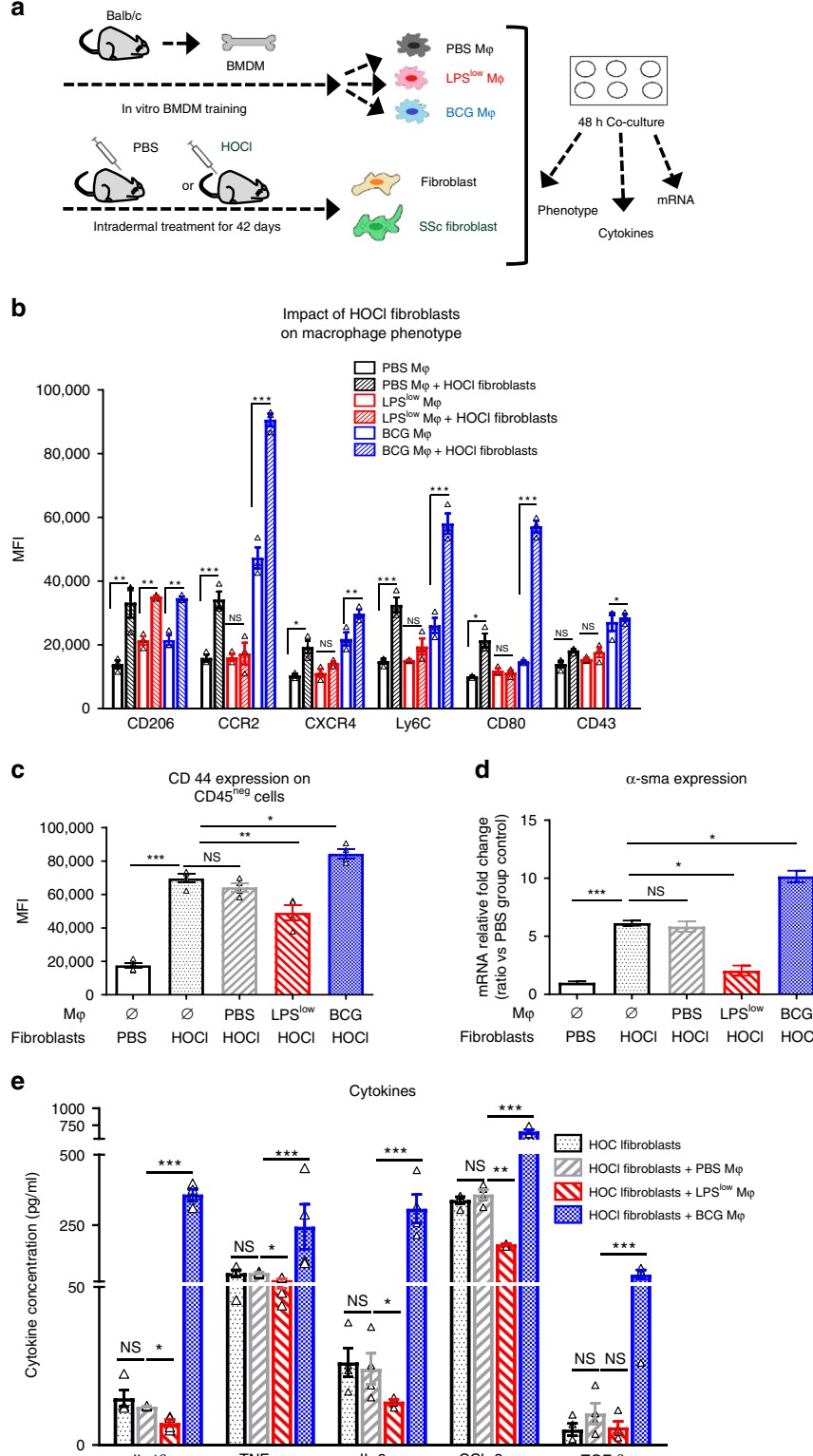

increased mortality to secondary infections[32]. Macrophages failure to produce pro-inflammatory cytokines and a shift toward elevated IL-10 secretion are believed to be the main implicated mechanisms for this observation. Lactate measurement in the supernatant of trained macrophages consolidated our results, as it has been demonstrated that cellular metabolism of macrophages undergo an important shift either to an increased glycolysis after BCG-training[9] or to a defect in energy metabolism in LPS-induced immuno-tolerance[15]. The dynamic changes in cytokines secretion by trained macrophages correlated with the H3K4me3 profile, a chromatin remodeling mark. Other histone modifications (H3K4 monomethylation, H3K27 acetylation, and H3K9 dimethylation) were previously described either for enhanced immune response or LPS tolerance[8,17,19] and should also be explored in the future in the context of inflammatory and autoimmune disease.

**Fig. 4 Dual interaction of trained macrophages and fibroblasts. a** Experimental setup of the co-culture model using trained bone-marrow derived macrophages (BMDM) and either resting dermal fibroblasts or activated dermal HOCl-fibroblasts. **b** Macrophage phenotype assessment by flow cytometry after 48 h of co-culture with activated HOCl-fibroblasts. Cellular pellet was harvested after 48 h of co-culture incubation. Cells were washed with warm PBS, gently detached from the plates, counted, and stained with antibody to perform flow cytometry. Macrophages were gated on CD11b and F4/80 positive cells among CD45 positive cell population and quantitative analysis of surface markers expression was performed. Data represent the mean fluorescence index (MFI) and SEM from the $n = 3$ biologically independent samples of each condition. **c** Fibroblasts were characterized among CD45 negative cells and quantitative expression of CD44 was assessed. Data represent the MFI and SEM from $n = 4$ biologically independent samples of each condition. **d** α-sma mRNA levels in cellular pellet. Results were expressed as fold increase versus the PBS-fibroblast condition, derived from $n = 4$ biologically independent samples (Unpaired $t$-test). **e** Cytokine production (pg/ml) was assessed by ELISA. Supernatant was collected after 48 h of co-culture incubation. Each box represents mean ± SEM from $n = 5$ biologically independent samples of each condition. The ANOVA test with Bonferroni correction was used to detect significant differences between the groups (unless stated). Mφ: Macrophages. NS: Not significant; *$p \leq 0.05$; **$p \leq 0.01$; ***$p \leq 0.001$. Source data are provided as a Source Data file.

The main goal of this study was to test the effects of trained macrophages on the outcome of inflammation-induced fibrosis. Indeed, fibrosis is frequently observed in the late stage of many chronic inflammatory diseases of infectious, autoimmune, allergic or metabolic origins and responsible for organ failure and mortality. Such phenomenon is observed in many connective tissue disorders and among those we choose SSc because of the extensive fibrosis of the skin and visceral organs that occurs in this chronic inflammatory autoimmune disease. Importantly, innate immune cells play a major role in SSc pathophysiology. Perivascular and dermal infiltration by mononuclear inflammatory cells is a hallmark of human SSc occurring at a very early step of the disease[33]. Then, macrophages rapidly adopt an aberrant activation state maintaining and reinforcing the inflammatory process that favors extensive fibrosis[24]. In our study, in vivo challenge of mice with BCG resulted in exacerbation of the disease, with an excessive fibro-inflammatory immune response upon HOCl injection. The excessive response consisted of an overproduction of pro-inflammatory cytokines including IL-1β and IL-17 that may activate fibroblasts to drive the synthesis of ECM resulting in progressive fibrosis[2,34]. The progressive fibrosis was also accompanied by an increased activation of B and T cells and elevation of IL-4 and IL-13 in conjunction with the shift towards Th2 cytokines observed in SSc[34,35].

Moreover, BCG-HOCl-mice showed early exacerbation of the inflammatory process including enhanced expression of chemokine receptors (CCR2 and CXCR4) and inflammatory markers (Ly6C) on dermal macrophages along with elevation of CCL2 production at sites of dermal lesions compared to HOCl-mice. Interestingly, human studies also report upregulation of these activation markers in dermal macrophages from SSc patients[36–38] and their increase are correlated with fibrosis progression in human and mice with SSc[39–46].

A major finding of our work is that LPS[low]-training induces a particular macrophages profile propitious to prevent exaggerated inflammation and the subsequent fibrosis. LPS[low]-trained animals showed an important reduction of their disease severity. LPS[low] training in vivo dampened inflammation with downregulation of inflammatory markers, chemokine receptors, and pro-inflammatory cytokines but enhanced the expression of iCOS ligand, DC-SIGN and IL-10 by dermal macrophages. Increased expression of iCOS ligand may participate to SSc amelioration. In a mouse model of bleomycin-induced SSc, iCOS-ligand deficiency has been shown to aggravate skin and lung fibrosis by regulating TGF-β induction[47]. Furthermore, upregulation of DC-SIGN expression on macrophages has been shown to promote tolerogenic effects in a mice transplantation model by inhibiting T cell activation[30] and to alleviate bleomycin-induced pulmonary fibrosis in mice[48]. Increased production of IL-10 in the skin of LPS[low] HOCl-mice may also contribute to reducing local inflammation and the fibrotic process. This result is in line with

the significant decrease of collagen production in the lung of mice with bleomycin-induced pulmonary fibrosis and treated by IL-10 gene therapy[49]. LPS and BCG signaling share the TLR4 receptor complex which drives the inflammatory responses through MAPK and NF-κB pathways[50,51]. In SSc, it has been clearly established that persistent TLR4-MD2 activation has a pathogenic role in fibrosis progression, with on overexpression on diseased fibroblasts[52,53]. LPS tolerance is accompanied by an impairment of the TLR4 signaling with an increase in the negative feedback regulators such as IRAK-M and A20[12,19]. BCG training has been shown to be dependent of the intracellular NOD2 receptor and the blockade of TLR2 and TLR4 did not abolish the training ability induced by BCG[10]. Thus it appears that our stimuli model uses a different signaling pathway that could contribute to explain the opposite effects obtained.

Collectively, this in vivo model of fibro-inflammatory disease demonstrates opposite immunomodulatory properties of LPS[low] and BCG in this context. LPS[low] training slows down the inflammation loop and prevents fibrosis through an induction of a "tolerant" phenotype and a shift in the cytokine production toward an anti-inflammatory balance while BCG training upregulates macrophages responsiveness to the inflammatory stimuli driven by HOCl injections and worsens the disease. The macrophage-(HOCl)-fibroblasts co-culture experiment, designed to investigate the immunomodulatory properties of trained macrophages, showed a bidirectional interaction between these cells. The myofibroblasts act as an inflammatory trigger to trained macrophages and shape their activation status and cytokine production. In return, these macrophages modulate the phenotype of myofibroblasts regarding the nature of their training, and behave like a potentiator (BCG) or an antagonist (LPS[low]) of the fibrotic process. In a similar approach, Song et al. showed that M2 macrophages increased the proliferation of fibroblasts and collagen production through the release of potent fibrogenic growth factors including TGF-β while M1 macrophages inhibited this fibrogenic process[54]. This study highlighted the effects of macrophages on fibroblast phenotype and fibrosis outcome but is based on in vitro differentiation of M1 and M2 macrophages. Thus, it may not reflect the more complex cytokine landscape that occurs in vivo upon macrophage training that combines both M1 and M2 features. Regarding how myofibroblasts can impact macrophages activation, it has been recently proposed that myofibroblasts can activate macrophages through IL-6 secretion and stimulation of STAT3 and Akt pathways[55].

Trained immunity can impact other cells than macrophages like NK cells[56] or innate lymphoid cells[57]. Therefore, injection of in vitro trained macrophages was performed to further ascertain their specific role in our system but also as a pre-clinical model of cellular therapy to dampen the fibro-inflammatory process in HOCl-mice. Previous studies have proposed macrophage therapy to overcome extensive fibrosis and promote scar resolution in

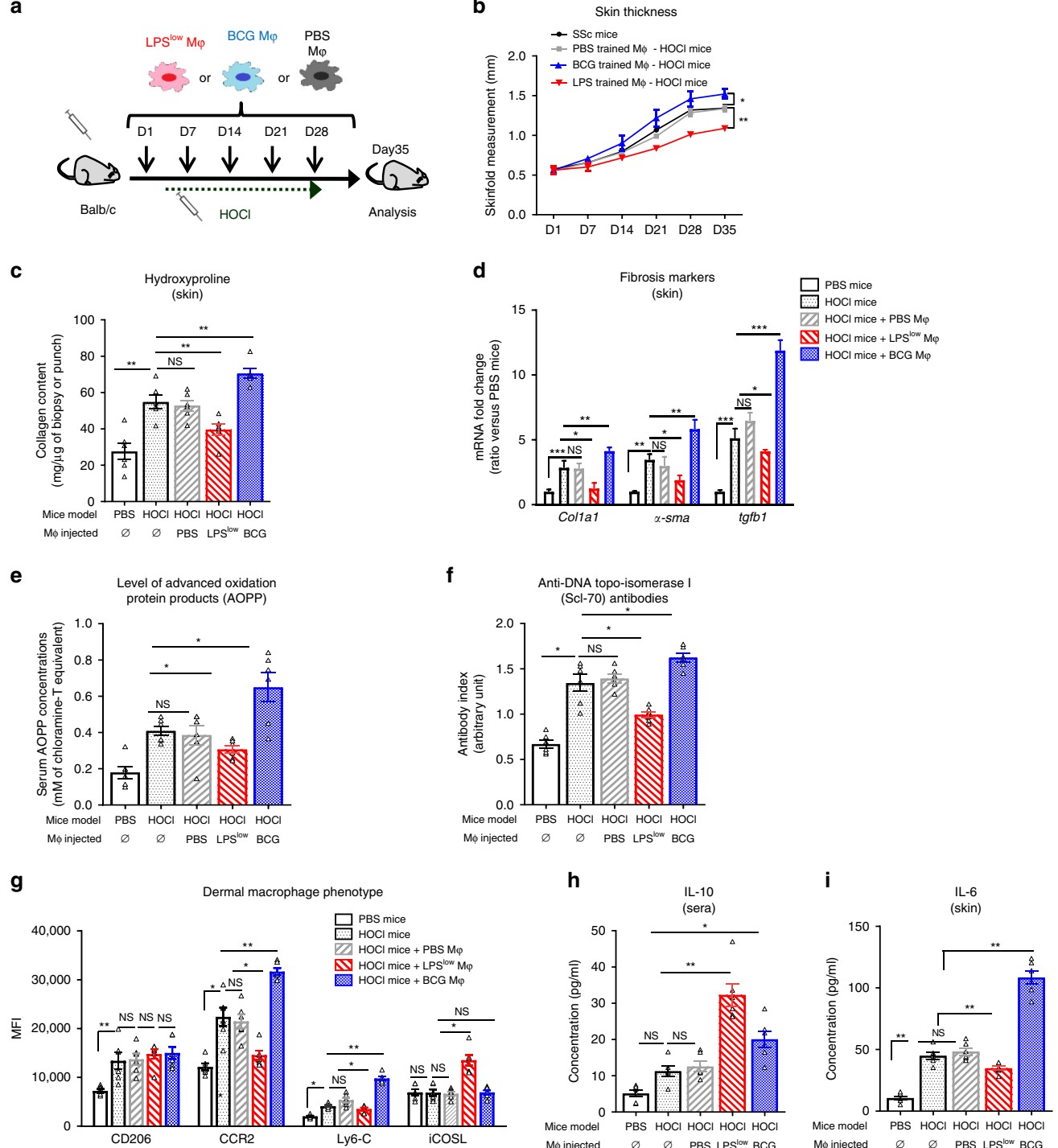

**Fig. 5 Adoptive transfer of LPS<sup>low</sup> trained macrophages ameliorates SSc. a** Experimental setup of therapeutic intraperitoneal injection of in vitro trained macrophages to HOCl-mice ($n = 6$ in each group). **b** Evolution of skin fold thickness in millimeters from days 1 to 35, measured weekly. **c** Collagen content in skin (Hydroxyproline dosage, mg/punch of biopsy). Each box represents mean ± SEM from $n = 6$ biologically independent mice. **d** Col1a1, α-sma, and Tgf-β mRNA levels in skin. Results were expressed as fold increase versus the PBS control group, derived from $n = 6$ biologically independent mice (Unpaired T-test). **e** Concentrations of anti-DNA topo-isomerase I (Scl70) antibodies in the sera from mice (Arbitrary Unit). Each box represents mean ± SEM from $n = 6$ biologically independent mice. **f** Concentration of advanced oxidation protein products (AOPP) in the sera from mice (mM of chloramine T equivalent). **g** Flow cytometric analysis of dermal macrophage phenotype at day 35. Dermal cell suspensions were prepared as for the previous experiments. Macrophages were gated on CD11b<sup>+</sup>F4/80<sup>+</sup>cells. Data represent the MFI and SEM from $n = 6$ biologically independent mice. **h, i** ELISA assessment of IL-6 production by skin-derived cells (**h**) and seric IL-10 concentration (**i**). Each box represents the mean concentration (pg/ml) ± SEM from $n = 6$ biologically independent mice. Statistics are shown between PBS and HOCl-mice first and then macrophage injected HOCl-mice (PBS, LPS<sup>low</sup> and BCG) versus un-injected HOCl-mice. The ANOVA test with Bonferroni correction was used to detect significant differences between the groups (unless stated). Mφ: Macrophages. NS: Not significant; *$p \leq 0.05$; **$p \leq 0.01$; ***$p \leq 0.001$. Source data are provided as a Source Data file.

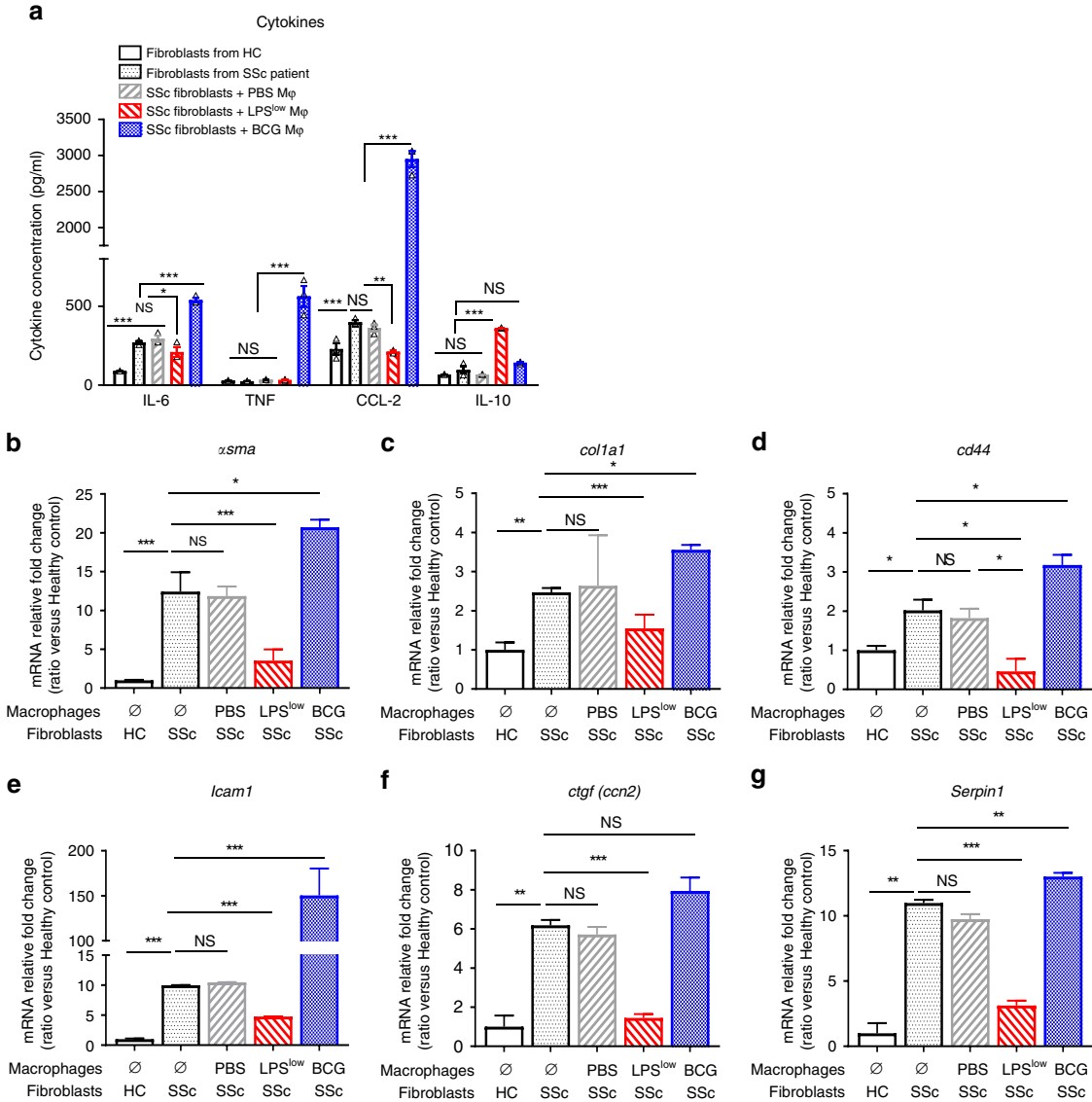

**Fig. 6 Effect of trained macrophages on fibroblasts in patients with SSc. a** Levels of IL-6, TNF, CCL-2, and IL-10 in co-culture supernatants. Supernatant was collected after 48 h of co-culture incubation. Cytokine production was assessed by ELISA (pg/mL). Each box represents mean ± SEM of $n = 3$ biologically independent samples. The ANOVA test with Bonferroni correction was used to detect significant differences between the groups. NS: Not significant; *$p \leq 0.05$; **$p \leq 0.01$; ***$p \leq 0.001$. **b–g** RT-qPCR assessment of α-sma, Col1a1, Cd44, Icam-1, Ctgf, and Serpin1 mRNA levels, after 48 h of co-culture. Results were expressed as fold increase versus the fibroblast from HC condition, derived from $n = 3$ biologically independent samples. Unpaired $T$-test was used to detect the significance. Mφ: Macrophages. NS: Not significant; *$p \leq 0.05$; **$p \leq 0.01$; ***$p \leq 0.001$. Source data are provided as a Source Data file.

liver cirrhosis models with promising results[58,59]. We showed that weekly injection of LPS[low] macrophages slows down the inflammatory process and the subsequent autoimmunity and fibrosis triggered by HOCl injections. Besides a reduced immune activation as the one observed in LPS[low] injected animals, we observed an increase in IL-10 level, a potent anti-inflammatory cytokine in the sera of mice injected with LPS[low]-macrophages. Similar observations have been made in a model of liver cirrhosis, where macrophages infusion induced an increase of the levels of the anti-inflammatory cytokine IL-10 associated with a reduction of the hepatic myofibroblasts population[60]. To gain further into the possible translation of this finding for clinical purpose, we designed a human co-culture model that revealed the potent effect that trained macrophages exert on diseased fibroblasts from SSc-patients. Most studies have analyzed the cross-talk between human SSc-fibroblasts with either T or B cells[61,62]. Data using

human macrophages are lacking. Results were consistent with those obtained in mice as LPS[low]-macrophages blocked the inflammatory process and reduced the fibrotic phenotype of diseased SSc-fibroblasts along with a drop in Icam1 expression. Interestingly, decreased levels of Icam-1 have been associated with reduced inflammatory cells infiltrates and attenuation of fibrosis in the tight-skin (TSK/+) mouse model of scleroderma[63].

In recent years, trained immunity has become a major field of interest as it shows possibilities in the design of promising tools of protection against re-infection and more efficient vaccine strategies. Our present study extends for the first time this concept to chronic fibro-inflammatory diseases. We provide for the first time evidences of depending on training procedure, trained macrophages can shape the immune responses involving T and B cell activation and cytokine production and can modulate fibroblasts transformation with a deep impact on the clinical evolution of

SSc. Our findings provide powerful insights on the roles of trained innate immunity in the pathophysiology of numerous chronic inflammatory diseases with fibrosis and organs failure and also a new possibility to potentially impact, through cellular therapy, the course of those frequent and severe affections. Indeed, in our study, we focused on the impact of trained immunity on the pathogenesis of SSc both in humans and mice as the prototypical of systemic fibro-inflammatory autoimmune diseases. However, our findings can be extended to other similar disorders where altered innate immune responses play a central role in triggering and maintaining the chronic inflammatory and autoimmune process such as rheumatoid arthritis, systemic lupus erythematous, type I diabetes, and multiple sclerosis[64–70]. Beyond these experimental models, our findings highlight for the first time how an anterior antigenic exposure caused by acute or chronic infections or vaccinations can, through innate immune training, shape the immune system towards an immunotolerant or an immunostimulatory status that could certainly impact both the onset and the course of inflammatory autoimmune diseases. This study not only highlights how trained immunity can be view as a new regulator of immune tolerance but also provides potential novel therapeutic tools for autoimmune and fibro-inflammatory diseases where trained immunity is inappropriately activated and could be adequately reprogrammed.

## Methods

**Reagents and chemicals**. Chloramine-T hydrate Ref # 857319, Trans-4-hydroxy-L Proline Ref # 56250, Potassium iodide Ref #60399, Potassium dihydrogen phosphate Ref #4,795488-Dimethylaminobenzaldehyde Ref #156477, Perchloric acid, 70% Ref # 24252 and 2-Methoxyethanol Ref#185469 were from Sigma-Aldrich (Saint Quentin Fallavier, France). LPS was from $E.\ Coli$ serotype 0127: B8 (Sigma Aldrich), BCG vaccine was purchased from Sanofi Pasteur and composed of 0.5 mg of the Brazilian strain (BCG Biomed-Lublin Laboratory).

**Mice**. Six-week-old female *BALB/c* mice weighing 16–20 g were purchased from Janvier Laboratory (Le Genest Saint Isle, France). All mice were housed in ventilated cages with sterile food and water ad libitum throughout the study. Animals received humane care in compliance with the guidelines implemented at our institution (INSERM and University Paris Descartes). The animal protocol used in this study was reviewed and approved by the local Ethic committee (Comité d'Ethide en matière d'Expérimentation animale Paris Descartes CEEA34 (Protocol CEEA34.CN.023.11).

**In vivo training of macrophages**. Three groups of 6 mice were used. Training of macrophages in vivo on *BALB/c* mice was achieved as follows (Fig. 1a): LPS training consisted of five peritoneal injections from day one to day five of a low dose of LPS (LPS$^{low}$) (0.1 mg/kg) to achieve macrophage immunoparalysis, as previously described[71]. BCG training was achieved by two peritoneal injection of BCG (37.5 µg/g corresponding to $6 \times 10^5$ CFU per mice per injection) on days 1–5, to induce a strong immuno-activated phenotype of macrophages as described[10]. The control group received two doses of peritoneal injection of 200 µL of PBS at day 0 and day 5. At day 7, all mice were killed. Macrophages were isolated from spleen (as described below) and were used to perform flow cytometry for phenotype assessment, chromatin immunoprecipation and cytokine production ex vivo upon challenge with an inflammatory dose of LPS (100 ng/ml)[72]

**Cells isolation and stimulation**. Spleen cells suspensions were prepared after hypotonic lysis of erythrocytes in potassium acetate solution and three washes in complete RPMI medium. For each mouse, splenocytes were enumerated using a Malassez counting chamber.
Selection of splenic macrophages was performed as described before[73]. FACS sorting was performed on splenic suspensions ($10^8$ cells per mouse) to generate F4/80 positive cells. The isolated cells were cultured for 24 h with complete RPMI supplemented with 10% FBS, 2 mM L-glutamine, 1% streptomycin-penicillin, 10 mM HEPES (all from Gibco Life Technologies) in 6-well plates (BD Bioscience) at 37 °C. The purity of the cells was checked by FACS analysis on randomized samples and exceeded 95% of CD11b$^+$ F4/80$^+$. For each mouse, a triplicate of $10^6$ of adherent splenic macrophages was then stimulated with 100 ng/ml of LPS for 24 h and supernatant was collected and stored at −80 °C.
Skin-derived immune cells isolation was achieved by collecting 8-mm-calibrated dermal punches of tissue were from the shaved back of each mouse and minced into small pieces. Tissue was then placed in a 60-mm petri dish containing 1 ml of dispase solution (Gibco, Ref #17703 Life Technologies) with 9 ml of complete RPMI medium and 1 mg/ml of collagenase (Sigma-Aldrich Ref #C2674-1G) and incubated for 3 h at 37 °C in an atmosphere of 5% $CO_2$. Cell suspension was then filtered in a 70-µm cell strainer, washed in complete RPMI medium and counted with Malassez counting chamber[74]. Skin immune cells stimulation was performed by 72 h of incubation at 37 °C in complete RPMI medium with 10 µg/ml of concanavalin A (Sigma Aldrich Ref # C5275). Then, supernatant was collected and stored at −80 °C.

**Flow cytometric analysis**. Splenocytes and skin-derived immune cells were prepared as described above. Cells suspensions were incubated for 20 min with 10 µg/mL anti-CD16/CD32 antibody (clone 93, eBiosciences, Catalog#14-0161-82, dilution 1/50) for Fc receptor saturation. Cells were then incubated with the appropriate labeled antibody at 4 °C for 30 min in the dark in PBS with 2% normal FBS. Flow cytometry was performed using a FACS Fortessa II flow cytometer (BD Biosciences), according to standard techniques. For the characterization of splenic, dermal and bone marrow derived macrophages, the monoclonal antibodies used were: F4/80-BV711 (BM8, Catalog#123147, dilution 1/200), CD11b-APC/Cy7 (M1/70, Catalog#101226, dilution 1/200), CD14-PE/Cy7 (Sa14-2, Catalog#123315, dilution 1/100), CD40-FITC (3/23, Catalog#124607, dilution 1/100), CD44-APC (IM7, Catalog#103012, dilution 1/200), CD69-PercP/Cy5.5 (H1.2F3, Catalog#104522, dilution 1/100), CD93-PercP/Cy5.5 (AA4.1, Catalog#136511, dilution 1/100), CXCR4-BV421 (L276F12, Catalog#146511, dilution 1/200), CD282-PE Toll-Like receptor 2 (TLR-2) (CB225, Catalog#148603, dilution 1/200), CD284-PE (TLR-4) (SA15-21, Catalog#145403, dilution 1/200), Ly6-C-PercP/Cy5.5 (HK1.4, Catalog#128011, dilution 1/200), CD275-PE (inducible T-cell COStimulator Ligand; iCOSL) (HK5.3, Catalog#107405, dilution 1/100), CCR2-PE (SA203G11, Catalog#150609, dilution 1/200), CD3-PE (145-2C11, Catalog#100308, dilution 1/200), CD4-APC-Cy7 (GK1.5 Catalog#100414, dilution 1/200), CD8-PE-Cy7 (53-6.7, Catalog#100722, dilution 1/200), CD19-Alexa Fluor 700 (6D5, Catalog#115528, dilution 1/150), CD45R/B220-PE (RA3-6B2, Catalog#103208, dilution 1/200) and CD80-APC (16-10A1, Catalog#104714, dilution 1/200) from BioLegend (Ozyme France, 78180 Montigny-le-Bretonneux) and CD45- FITC (RA3-6B2, Catalog#11-0452-82, dilution 1/200), CD11b-BV510 (M1/70, Catalog#562950, dilution 1/200), CD80-FITC (16-10A1, Catalog#553768, dilution 1/200), CD209-APC (DC-SIGN) (LWC06, Catalog#17-2092-80, dilution 1/200), CD43-BV421 (S7, Catalog#562958, dilution 1/200), CD206-Alexa Fluor 647 (MR5D3, Catalog#565250, dilution 1/200), IA-IE-FITC (MHC-II) (2G9, Catalog#553623, dilution 1/100) from eBiosciences (Thermo Fisher Scientific, Villebon-Sur-Yvette, France). Data were analyzed with FlowJo software (Tree Star, Ashland, OR). Gating strategies used for flow cytometry analysis is shown in Supplementary information, Fig. 10.

**ELISA cytokine detection**. Assessment of IL-1β (Mouse Ref # 88–7013, Human Ref # 88-7261-88), IL-4 (Mouse Ref # 88-7044-88), IL-6 (Mouse Ref # 88-7064-88, Human Ref # 88-7066-88), IL-10 (Mouse Ref # 88-7105-88, Human Ref # 88-7106-88), IL-13 (Mouse Ref # 88-7137-88), IL-17 (Mouse Ref # 88-8711-88), TNF-α (Mouse Ref # 88-7324-88, Human Ref # 88-7346-88), CCL2 (Mouse Ref # 88-7391-88, Human Ref # 88-7399-88), and TGF-β(Mouse Ref # 88-8350-88) was performed by specific mouse ELISA kits from eBiosciences (Invitrogen) (Thermo Fisher Scientific, Villebon-Sur-Yvette, France). Concentrations were calculated from a standard curve according to the manufacturer's protocol.

**Lactate measurement**. Lactate release in the supernatant of trained mice splenic macrophages stimulated with inflammatory dose of LPS was measured using the Cobas® 8000 modular analyzer series—Diagnostics Roche, France.

**Chromatin immunoprecipitation and quantitative PCR**. Adherent splenic macrophages were isolated as described above and fixed in 37% methanol-free formaldehyde. The cells were then centrifuged and the pellet was stored at −80 °C. The chromatin preparation and immunoprecipitation was performed as described in the Diagenode user manual (Ref # C01010059). A total of 300 µL of chromatin was prepared from $10^6$ frozen fixed cells. The sonication was performed on a Bioruptor® Pico, for 10 cycles (30 s ON, 30 s OFF). The immunoprecipitation was performed with 100 µL of the obtained chromatin in a 300-µL final volume with H3K4me3 antibody and IgG as negative control. The immunoprecipitated chromatin and 10 µL of control input chromatin were decrosslinked and resuspended in 40 µL. In all, 2 µL of this DNA were then used in a 10-µL final volume for qPCR (using SensiAST™ SYBR® No-ROX Kit from Bioline) and the appropriate primers (chosen in the H3K4me3 mouse ChIP seq peaks from data of Quintin et al.[8]. As there was no H3K4me3 peak in the vicinity of IL-1β gene according to the CHiP-seq data from published data[17,75], we did not perform ChIP-qPCR at this locus. The primers are listed in Supplementary Table 1.

**Induction of mice systemic fibrosis**. To address the impact of immune training on the fibro-inflammatory affection, we used a well-established mouse model of systemic fibrosis based on daily intradermal injections of the reactive oxygen species hypochlorous acid (HOCl). This model mimics the inflammatory, fibrotic, and autoimmune features of the human disease systemic sclerosis (SSc)[76]. Mice were randomly distributed into experimental and control groups (12 mice per

group). SSc was induced according to the protocol described by Kavian et al.[77]. A total of 200 μL of a hypochlorous acid (HOCl) solution was prepared extemporaneously and injected intradermally into the shaved backs of mice (one injection of 100 μl in each flank) using a 27-gauge needle, every day for 6 weeks (SSc-mice). HOCl was prepared by adding 80 μL of a NaClO (SAS Richet 02 Javel) solution (9.6% active chlorine) to 280 μL of $KH_2PO_4$ (pH 6.2)[78].

**Skin and lung fibrosis assessment.** Skin fibrosis was assessed by measuring the dermal thickness of the shaved backs of mice. This was determined by double-blinded scientists with a caliper and expressed in millimeters once a week until the end of the experiment. Collagen content in the skin and in the lungs was assessed by the hydroxyproline content evaluation as recommended by Woessener[79]. Briefly, after digestion of punch biopsies (5-mm diameter) in HCl (6 M) for 3 h at 120 °C, the pH of the samples was adjusted to 7. Samples were then mixed with chloramine T (0.06 M) and incubated for 20 min at room temperature. Perchloric acid (3.15 M) and p-dimethylaminobenzaldehyde (20%) were then added and samples were incubated for an additional 20 min at 60 °C. The absorbance was determined at 557 nm with a microplate spectrophotometer (Fusion; PerkinElmer, Wellesley, MA).

**Assays of anti-DNA topoisomerase I autoantibodies in sera.** Levels of anti-DNA topoisomerase I IgG antibodies were detected using the Scl-70 IgG ELISA Kit. Diluted (1/4) mouse serum was distributed into the wells coated with purified calf thymus DNA topoisomerase I of a microtiter plate (Abnova, Taipei City, Taiwan). The conjugated anti-mice Ig horseradish peroxidase (Dako, Glostrup, Denmark) secondary antibody (1/100) was then added and the absorbance was read at 450 nm. Results were expressed as antibody index using the cut-off value derived from of the Calibrator optic density.

**Assays of advanced oxidation protein products in sera.** Sera were diluted (1:5) in PBS and distributed (200 μL) onto a 96-well plate with 10 μL of 1.16 M potassium iodide. Calibration used a two-fold dilution series of chloramine-T solution within the range from 0 to 100 mM. The absorbance was read at 340 nm on a microplate reader (Fusion; PerkinElmer) and AOPP concentration was expressed as mM of chloramine-T equivalents[80].

**Bone marrow derived macrophage isolation and culture.** Bone marrow cells (BMCs) were collected from femurs and tibias of female *BALB/c* mice and differentiated into Bone Marrow Derived Macrophages (BMDM) using BMDM medium (RPMI with 20% L929 cell supernatant containing GM-CSF required to macrophage differentiation, 10% heat inactivated Fetal Bovine Serum (FBS), 1% antibiotic/antimycotic, 1% HEPES, and 1% sodium pyruvate) (Eurobio Ingen. Les Ulis, France). At day 7, cells were removed from plastic dishes by incubation in $Ca^{2+}/Mg^{2+}$-free PBS and collected by brief trypsin-EDTA (Gibco) treatment. Cells were spun at 1200 rpm for 7 min to form a pellet and then re suspended in complete RPMI media[81,82].

**Training of bone marrow derived macrophages.** For in vitro BCG treatment, dry BCG vaccine (BCG Biomed-Lublin Laboratory) was diluted in sterile PBS and added to the plate wells containing 10 million BMDM at a concentration of 1 bacillus for 1 cell and incubated for 24 h[83]. As for LPS pretreatment there are several controversies on the doses and the duration of in vivo in published observations[19,84], leading to opposite effects (priming for super low doses and blocking for high doses), we performed a dose-response adjustment (Supplementary Fig. 1) based on numerous experimental studies[14,19,85]. To achieve macrophage LPS training, one daily dose of 10 ng/mL (Referred as low dose) of LPS from *E. Coli* serotype 0127:B8 (Sigma Aldrich), was added from days 1 to 3 with daily change of culture medium to avoid cumulative doses toxicity. Control macrophages were obtained by daily treatment with 100 μL of sterile PBS. All cells were cultured in complete DMEM media with 10% FCS and incubated at +37 °C in an atmosphere containing 5% $CO_2$. Cells were then collected by brief trypsin-EDTA (Gibco) treatment, counted and used either for co-culture experiment or intraperitoneal injection.

**Isolation of skin fibroblasts.** Dermal tissue explants were harvested from shaved backs of mice after 6 weeks of HOCl treatment (to obtain activated myofibroblasts culture) or from 6 weeks of PBS treatment (to obtain control inactivated fibroblasts) and minced into small pieces, followed by digestion with 1 mg/ml collagenase (Sigma-Aldrich Ref C2674) and 0.5 mg/ml of dispase (Liver Digest Medium, Gibco) for 4 h at 37 °C. The cell suspension was then filtered in a 100-μm cell strainer, washed three times in DMEM high glucose (Gibco) and cultured in the presence of DMEM supplemented with 10% FBS, 1% streptomycin-penicillin, and 1% of Ciprofloxacin (Panpharma 2 mg/ml). After ~1 week incubation at 37 °C in an atmosphere of 5% $CO_2$, fibroblasts were detached by brief trypsin-EDTA (Gibco) treatment and resuspended in DMEM. Collected fibroblast were counted and used for co-culture experiments.

**Histopathological analysis.** Skin and lung biopsies were fixed with 10% formaldehyde and set in paraffin. Serial 6-μm sections were prepared and stained with Picrosirius Red. Slides were examined by standard brightfield microscopy (Olympus BX60, Rungis, France) by a pathologist who was blinded to the experimental group assignment.

**Mouse BMDM–dermal fibroblast co-culture.** Fibroblasts, activated fibroblasts (SSc-fibroblasts) and macrophages were seeded at a density of $5 \times 10^4$ cells/well onto the surface of a 6-well tissue culture plate at different conditions as follow: fibroblasts were seeded alone (in DMEM supplemented with 10% FBS, 1% streptomycin-penicillin and 1% of Ciprofloxacin) or in combination with macrophages trained with either PBS, LPS$^{low}$, or BCG (as described above) in complete RPMI medium enriched with 1 mM of sodium pyruvate (Gibco Ref #11360070). The same cellular combination was also performed for the co-culture of activated myofibroblasts. Trained macrophages were also cultivated alone in complete RPMI medium to evaluate the basal activation status. All the co-cultures conditions were performed in sixplicate. Cells were then incubated for 48 h at 37 °C in 5% $CO_2$. Supernatants were collected and stored at −80 °C for further cytokines measurement. Cellular pellets were used to perform flow cytometric characterization or mRNA extraction for RT-qPCR analysis.

**Quantitative reverse transcription PCR.** Total mRNA was extracted from crushed samples with TRIzol reagent (Invitrogen). One step RT-qPCR was performed using QuantiTect SYBR® Green RT-PCR Kit on a LightCycler 480 II instrument (Roche Applied Science, France). RT-qPCR was carried out for 45 cycles, with a denaturing phase of 15 s at 94 °C, an annealing phase of 30 s at 60 °C, and an extension of 30 s at 72 °C. Samples were normalized to mRNA expression of housekeeping genes (*β-actine* for mice samples and *GAPDH* for human samples), and results were expressed as fold increase using the formula 2−^^Ct. Primers used for PCR are listed in Supplementary Table 1.

**Adoptive transfer of trained macrophages in HOCl-mice.** *BALB/c* mice were separated into 4 groups containing 6 mice each: HOCl-mice received weekly intraperitoneal injection of 150 μl sterile PBS, HOCl-LPS$^{low}$-mice received weekly injection of 1.5 million LPS$^{low}$-treated macrophages in 150 μl of sterile PBS, BCG-HOCl-mice received weekly injection of 1.5 million BCG-treated macrophages in 150 μl of sterile PBS and PBS-HOCl-mice received weekly injection of 1.5 million PBS-treated macrophages in 150 μl of sterile PBS. All groups had a daily intradermal injection of 200 μl of HOCl-generating reagents to induce SSc. After 35 days the animals were killed and analysis of the skin, spleen, and lungs were performed.

**Human fibroblast-macrophages co-culture experiments.** Blood samples were obtained from healthy volunteers who gave their informed consent, and PBMC were isolated from buffy coat on a Ficoll density gradient. Cells were washed twice, adjusted to $10^6$ cells per well in a 6-well plate and incubated in complete RPMI medium for 12 h at 37 °C in 5% $CO_2$. Adherent monocytes were selected by washing out non adherent cells with warm PBS and FACS sorting of CD14 positive cells was performed with a purity of ≈94%. Cells were then counted and trained with LPS, BCG, or PBS as described above. Human diseased fibroblasts were obtained from 4 millimeters dermal biopsies from the forearm skin with active fibrotic lesions from a 53-year-old and a 48-year-old patients with a diffuse form of systemic sclerosis diagnosed in 2017 and 2013 respectively, fulfilling the ACR criteria for SSc[86] and without ongoing immunosuppressive treatment. Normal fibroblasts were collected from a dermal biopsy of a 45-year-old healthy control. Skin biopsies derived from 2 SSc patients and 2 healthy subjects were kindly provided by Pr. Yannick Allanore, Rheumatology Department, Cochin Hospital, Paris, France. Samples were immediately diced with scalpels in collagenase for 2 h at 37 °C. Cells were rinsed, filtered, and cultured in complete DMEM at 37 °C in 5% $CO_2$. Fibroblasts were observed after 3–5 days and expanded. Fibroblasts were then seeded in 6-well-plates ($5 \times 10^4$ cells) at the same different combinations of the mice co-culture model described above. All the co-cultures conditions were performed in triplicate. Cells were incubated for 48 h at 37 °C in 5% $CO_2$. Supernatant was collected and stored at −80 °C for further cytokines measurement. Cellular pellet was used to perform mRNA extraction for RT-qPCR analysis. The local ethical committee of Cochin Hospital and the local institutional review boards [CCP (Comité de Protection des Personnes) Paris Ile De France3] approved the study, and all subjects provided written informed consent.

**Statistical analysis.** All quantitative data were expressed as means ± SEM. Groups were compared using one-way ANOVA followed by a Dunn multiple comparison test. Mann–Whitney test was used to compare two groups. All analyses were carried out using the GraphPad Prism 5.0 statistical software package (San Diego, CA), and $p$ values < 0.05 were considered significant.

**Reporting summary.** Further information on research design is available in the Nature Research Reporting Summary linked to this article.

## Data availability

All data generated or analyzed during this study are included in the manuscript and its Supplementary files or are available from the authors upon reasonable request. Raw data are also provided in the Source Data file. For further request please contact the corresponding author.

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

## Acknowledgements

We would like to thank the Plateforme of Cytometry and Immuno-biology CYBIO of Cochin Institute, Paris, for flow cytometry and data analysis. This work was supported by grants from University Paris Descartes and INSERM organization.

## Author contributions

Conceptualization by F.B. and N.K.; Methodology by M.J., L.G.C.R, Q.D. and C.N.; Investigation by M.J., C.C., L.G.C.R. and S.C.; Formal analysis by M.J., C.C., C.N., and L.D. Writing –Original Draft by M.J., F.B. and N.K.; Writing –Review & Editing by M.J., L.D., Y.A., F.B. and N.K.; Funding Acquisition by F.B. and N.K.; Supervision by F.B., C.N. and N.K.

## Competing interests

The authors declare no competing interests.
