## [Peer Review File · Nature Communications]

Reviewers' comments:

Reviewer #1 (Remarks to the Author):

The manuscript by Jeljeli and colleagues investigates the immunomodulatory properties of trained immunity on the outcome of fibrosis in diseases with autoimmune traits. The subject of the study has a high relevance both for understanding an aspect of innate immune memory which has not been investigated until now, as well for providing new insights in the pathophysiology of fibrotic diseases. The study is well performed and the manuscript is clearly written. The authors should be also commended by combining experimental studies with studies relevant in patients with systemic sclerosis.

Comments:

1. The Abstract is not very clearly written. More details regarding the precise model/disease investigated needs to be given, in order for the reader to understand exactly what has been done, and the relevance for systemic sclerosis patients.
2. The phenotype of trained immunity in macrophages has been described to be associated with increased glycolysis. Is that the case also in the model described here by the authors? This can be easily assessed by measuring lactate in the supernatants of the LPS re-stimulated trained spleen macrophages.
3. In Fig.1 the panel numbers in the figure and in the legend do not match with each other.
4. H3K4me3 data supports the hypothesis of the authors that epigenetic reprogramming forms the molecular substrate of trained immunity of macrophages in the model described. However, the authors should discuss and acknowledge that this is only one of the epigenetic marks influencing gene transcription, and future studies should address the epigenetic mechanisms of this process in more detail, including study of other histone marks (H3K4me1, H3K27Ac, H3K9me2).
5. In the Discussion, it would be useful if the authors would discuss a bit more of the types of human pathologies in which their findings are relevant. At this moment the Discussion mainly focuses on the experimental models used, but a broader discussion of the implications of the study is missing.

Reviewer #2 (Remarks to the Author):

The manuscript examines the effect of pre-stimulating (training) macrophages prior to induction of HOCl fibrosis. The ideas are presented in a novel way and effects are seen on HOCl murine fibrosis.

COMMENTS:

I don't understand what the authors mean by " Regarding systemic markers of SSc spreading." They are referring I think simply to markers of the disease, but do they mean human or murine disease? Anti-topoisomerase is of course a marker of the human disease but is "advanced oxidation protein products"? If the authors are referring to markers of their model, then I would state that these are markers of progressive HOCl-induced disease or similar.

I am (barely) willing to accept the abbreviation of SSc-PBS for their HOCl treated mice (it would be better to refer to them by HOCl-PBS), but sentences like "Thus, in vivo training of mice with LPSlow prior to the induction of systemic sclerosis...are really not acceptable. Confusing HOCl-induced murine disease with SSc is confusing, particularly as there is no evidence that human SSc is caused by HOCl.

In the section related to Figure 4, the authors should refer to the adherent cells from HOCl treated mice as HOCl fibroblasts and the authors haven't shown that these are myofibroblasts (stained

them with alpha-SMA), even though I see the SMA mRNA expression is higher. The possibility that the macrophage training is affecting myofibroblast differentiation in the discussion based on these observations is fine.

All of these apparently picayune concerns about how the authors describe the cells from the HOCl mice becomes even more important for distinguishing the results with cells that are actually coming from SSc patients in the last section. However, even here the authors should not refer to the cell cultures as myofibroblasts, as studies have shown that myofibroblasts represent only a fraction and are frequently the minority of cells from SSc skin biopsies. There is an exhaustive controversial literature regarding whether fibroblasts from SSc patients are different from control fibroblasts (and this is not particularly important for the impact of these results in this manuscript).

Regarding Figure 6, there are much more interesting markers of TGF β activation and/or fibrosis they might have examined than CD44 and ICAM, such as CTGF or SERPINE1.

The discussion should compare the anticipated signaling of BCG to LPS. The latter is relatively understood and reference(s) regarding the role of TLR4 in SSc pathogenesis should be cited and mentioned briefly. As BCG also activates TLR4 but has other receptors (TLR2), this should also be considered, i.e., why are they seeing different results.

Response to referees

Reviewers' comments:

Reviewer #1 (Remarks to the Author):

The manuscript by Jeljeli and colleagues investigates the immunomodulatory properties of trained immunity on the outcome of fibrosis in diseases with autoimmune traits. The subject of the study has a high relevance both for understanding an aspect of innate immune memory which has not been investigated until now, as well for providing new insights in the pathophysiology of fibrotic diseases. The study is well performed and the manuscript is clearly written. The authors should be also commended by combining experimental studies with studies relevant in patients with systemic sclerosis.

We thank the referee #1 for the interest he showed in our work that aimed to investigate a new aspect of innate immune memory and its potential clinical benefits in fibro-inflammatory auto-immune diseases. Point-by-point responses to the helpful comments are given below. The indications of lines and pages are referred to the marked manuscript.

Comments:

1. The Abstract is not very clearly written. More details regarding the precise model/disease investigated needs to be given, in order for the reader to understand exactly what has been done, and the relevance for systemic sclerosis patients.

We agree with the referee and added more details about the murine model investigated and the results obtained for a better understanding. We have re-written the Abstract (Page2) and raised the clinical relevance of the paper for systemic sclerosis patients.

2. The phenotype of trained immunity in macrophages has been described to be associated with increased glycolysis. Is that the case also in the model described here by the authors? This can be easily assessed by measuring lactate in the supernatants of the LPS re-stimulated trained spleen macrophages.

The authors thank the referee #1 for this experimental suggestion that allowed us to consolidate our *in vitro* model with the metabolic modifications that characterize innate immune training and to add this important data to the one we already gained (phenotypes, cytokine production and histone H3K4 trimethylation).

As suggested, we performed a lactate measurement using the Cobas® 8000 modular analyzer series - Diagnostics Roche on both supernatants from LPS re-stimulated trained spleen macrophages (Experience 1 from Fig. 1g) and supernatants from LPS re-stimulated PBMCS of healthy controls used for the human co-culture experience (from Fig 6g).

Lactate release from BCG-trained murine spleen macrophages were significantly increased when compared to the untrained macrophages while LPS^{low} training induced a reduced production of lactate (BCG trained mice: 23.55 ± 0.72 mmol/l vs 20.3 ± 0.62 mmol/l, $p=0.002$ and LPS^{low} trained mice: 17.35 ± 0.85 mmol/l vs 20.3 ± 0.62 mmol/l, $p=0.01$; see Fig 1g).

The same tendency on supernatants from trained human macrophages re-stimulated with inflammatory dose of LPS was also observed. (BCG trained macrophages: 1.87 ± 0.03 mmol/l vs 1.72 ± 0.03 mmol/l, $p=0.01$ and LPS^{low} trained macrophages: 1.64 ± 0.03 mmol/l vs 1.72 ± 0.03 mmol/l, $p=0.09$). Data not shown (for referees only).

These results are consistent with previous observation describing the metabolic reprogramming occurring during BCG-induced trained immunity ¹ and LPS-induced immunotolerance ².

Method of lactate measurement was included in the Materials and Methods sections (Page 8, line 167-170):

“Lactate measurement

Lactate release in the supernatant of trained murine splenic macrophage stimulated with inflammatory dose of LPS was measured using the Cobas® 8000 modular analyzer series - Diagnostics Roche, France.”

The results have been added to the Result section (Page 15, Lines 352-359) as follows:

The reprogramming of cellular metabolism is a hallmark of trained immunity. The phenotype of trained macrophages has been described to be associated with increased glycolysis or a defect in the energy metabolism ¹⁻³. We evaluated the lactate production that reflects glycolysis activity, in the supernatant of trained splenic macrophages stimulated with LPS^{high}. As shown in Fig. 1g, lactate release from BCG-trained murine spleen macrophages were significantly increased compared to the untrained macrophages ($p=0.002$) while LPS^{low} training induced a reduced production of lactate ($p=0.01$).

Also we have added in the discussion section the following paragraph (Page 32 Lines 697-701):

Lactate measurement in the supernatant of trained macrophages consolidated our results, as it has been demonstrated that cellular metabolism of macrophages undergo an important shift either to an increased glycolysis after BCG-training ¹ or to a defect in energy metabolism in LPS-induced immuno-tolerance ².

3. In Fig.1 the panel numbers in the figure and in the legend do not match with each other.

We apologize for this mistake. We have corrected the panel number in Fig1 to match with the legend (Pages 14 and 15).

4. H3K4me3 data supports the hypothesis of the authors that epigenetic reprogramming forms the molecular substrate of trained immunity of macrophages in the model described. However, the authors should discuss and acknowledge that this is only one of the epigenetic marks influencing gene transcription, and future studies should address the epigenetic mechanisms of this process in more detail, including study of other histone marks (H3K4me1, H3K27Ac, H3K9me2).

We fully agree with this comment and we now mention these other epigenetic modifications in the introduction of the new version of the manuscript (Page 4, Lines 80-82):“chromatin modifications (H3K4me1, H3K27Ac, H3K9me2, H3K4me3)^{1,4-6} on promoters of genes encoding cytokines, signaling proteins and surface markers”.

We also have indicated in the discussion section the importance to study these epigenetic marks in the future in the context of autoimmunity (Page 32, Lines 702-705).

“ Other histone modifications (H3K4 monomethylation, H3K27 acetylation, H3K9 dimethylation) were previously described either for enhanced immune response or LPS tolerance and should also be explored in the future in the context of inflammatory and autoimmune disease”.

5. In the Discussion, it would be useful if the authors would discuss a bit more of the types of human pathologies in which their findings are relevant. At this moment the Discussion mainly focuses on the experimental models used, but a broader discussion of the implications of the study is missing.

We agree with the reviewer’s comment and we have modified the discussion in order to highlight the clinical relevance of our study. Indeed, innate immunity plays a central role in several human autoimmune and inflammatory diseases. It is clear that these diseases could benefit from the immunomodulatory properties of innate memory. In our study we focused on the impact of trained immunity on the pathogenesis of systemic sclerosis, a prototypical autoimmune fibro-inflammatory disease. However, we believe that our findings can be extended to other autoimmune and auto-inflammatory diseases where altered innate immune response play a central role in triggering and maintaining the chronic inflammatory process such as rheumatoid arthritis, systemic lupus erythematosus, type I diabetes, and multiple sclerosis⁷⁻¹³

We have added the following paragraph in the discussion section, as follow (Page 36, Lines 808-822)

“Indeed, in our study, we focused on the impact of trained immunity on the pathogenesis of systemic sclerosis both in humans and mice as the prototypical of systemic fibro-inflammatory autoimmune diseases. However, our findings can be extended to other autoimmune and auto-inflammatory diseases where altered innate immune responses play a central role in triggering and maintaining the chronic inflammatory and autoimmune process such as rheumatoid arthritis, systemic lupus erythematosus, type I diabetes, and multiple

sclerosis⁷⁻¹³. Beyond these experimental models, our findings highlight for the first time how an anterior antigenic exposure caused by acute or chronic infections or vaccinations can, through the particular training of innate immune cells, shape the immune system towards an immunotolerant or an immunostimulatory state that could certainly impact both the onset and the course of many inflammatory autoimmune diseases. This study not only highlights how trained immunity can be viewed as a new regulator of immune tolerance but also provides potential novel therapeutic tools for autoimmune and auto-inflammatory diseases where trained immunity is inappropriately activated and could be adequately reprogrammed.

Reviewer #2 (Remarks to the Author):

The manuscript examines the effect of pre-stimulating (training) macrophages prior to induction of HOCl fibrosis. The ideas are presented in a novel way and effects are seen on HOCl murine fibrosis.

We kindly thank the referee #2 for his interest in our work that aimed to apply the immunomodulatory properties of innate memory by macrophage training in the murine model mimicking the human systemic sclerosis.

COMMENTS:

1- I don't understand what the authors mean by "Regarding systemic markers of SSc spreading" They are referring I think simply to markers of the disease, but do they mean human or murine disease? Anti-topoisomerase is of course a marker of the human disease but is "advanced oxidation protein products"? If the authors are referring to markers of their model, then I would state that these are markers of progressive HOCl-induced disease or similar.

There are 2 forms of SSc in humans, the limited and the diffuse forms. In the limited form of SSc, we generally observe skin fibrosis limited to the hands, forearms, feet and lower legs, and an absence of lung fibrosis¹⁴. In the diffuse form of the disease, patients generally present with an extended skin fibrosis along with a lung fibrotic disease.

We have shown in our paper describing the HOCl-induced SSc murine model that HOCl injection in the dermis specifically induces a form of disease resembling diffuse SSc, presenting with skin fibrosis but more importantly with lung involvement along with kidney vascular alterations¹⁵. In this paper, we also show that other forms of ROS can induce a different disease phenotype. For instance peroxynitrites (ONOO⁻) induce a fibrosis that is limited to the skin only, without any visceral organ involvement as observed in the limited form of SSc.

In the same paper, we showed that sera from HOCl-SSc mice and of patients with diffuse SSc contains high levels of Advanced Oxidation protein products (AOPP), and that the latter can trigger fibroblast hyper-proliferation and endothelial H₂O₂ hyper-production, thus allowing the systemic spreading of the disease from skin (site of HOCl exposure in mice, and site of ROS exposure possibly in humans too) to internal organs including lungs (see figure 4 and 6 for Reviewers only from Servettaz et al. Journal of Immunology 2009¹⁵). Therefore, AOPP mediate the diffusion of the disease from the first rapid skin involvement (due to intra-dermal injections of HOCl) to other visceral organs including lungs. We have also shown that the immune system plays a synergistic role with AOPP to trigger disease propagation, as SCID-mice injected with HOCl only skin fibrosis with a significant reduction of lung fibrosis compared to WT-mice.

AOPP are a marker of systemic oxidative stress^{16,17} and others have also demonstrated in some clinical studies that patients with SSc have elevated levels of AOPP¹⁸. In this latter study forty SSc patients were compared with 20 matched healthy subjects. The median concentrations of AOPP among other markers of SSc vascular damage VEGF, sVEGFR-1, sVCAM-1, and carbonyl residues, were found significantly higher in SSc patients than in healthy subjects at baseline.

We understand that the initial wording «Regarding systemic markers of SSc spreading» may not be very clear, and we have replaced it in the new version of the paper by the following sentence (Page 21, Lines 469-473):

“Beyond markers of fibrosis, the levels of AOPP and anti-DNA-topoisomerase I autoantibodies that are systemic markers of the diffuse form of the disease, tended to be elevated in HOCl-BCG mice (p=0.09 and p=0.01 respectively, Fig. 3f and 3g) while HOCl-LPS^{low} mice displayed significant reduced levels of those markers in the blood compared to HOCl-PBS mice (p=0.03 and p=0.01, respectively, Fig. 3f and 3g).”

2- I am (barely) willing to accept the abbreviation of SSc-PBS for their HOCl treated mice (it would be better to refer to them by HOCl-PBS), but sentences like "Thus, in vivo training of mice with LPS^{low} prior to the induction of systemic sclerosis...are really not acceptable. Confusing HOCl-induced murine disease with SSc is confusing, particularly as there is no evidence that human SSc is caused by HOCl.

We agree with the reviewer's comment, and understand the confusion between the HOCl-murine disease and the human-SSc.

We and others have shown that there is evidence that oxidative stress characterizes the pathophysiology of the initiation and development systemic sclerosis, and an increasing number of data and extensive studies report an imbalanced oxidative stress in patients with SSc¹⁵⁻¹⁹. Redox signaling plays a well-described role in the pathogenesis of the fibrosis, vascular but also immunological phenomena in SSc. Although we agree with the reviewer that the identification of the primary cause leading to ROS production in human patients is still lacking, and that there is no direct evidence that HOCl is the cause of SSc. To avoid any confusion and as suggested by the reviewer, we have changed the abbreviation “SSc-PBS” by

“HOCl-PBS”, and we have changed the sentence (Page 22 Lines 473-474) “Thus, in vivo training of mice with LPS^{low} prior to the induction of systemic sclerosis” by “Thus, in vivo training of mice with LPS^{low} prior to the induction of the disease by HOCl-injections”

In the section related to Figure 4, the authors should refer to the adherent cells from HOCl treated mice as HOCl fibroblasts and the authors haven't shown that these are myofibroblasts (stained them with alpha-SMA), even though I see the SMA mRNA expression is higher. The possibility that the macrophage training is affecting myofibroblast differentiation in the discussion based on these observations is fine.

All of these apparently picayune concerns about how the authors describe the cells from the HOCl mice become even more important for distinguishing the results with cells that are actually coming from SSc patients in the last section. However, even here the authors should not refer to the cell cultures as myofibroblasts, as studies have shown that myofibroblasts represent only a fraction and are frequency the minority of cells from SSc skin biopsies. There is an exhaustive controversial literature regarding whether fibroblasts from SSc patients are different from control fibroblasts (and this is not particularly important for the impact of these results in this manuscript).

Regarding Figure 6, there are much more interesting markers of TGF β activation and/or fibrosis they might have examined than CD44 and ICAM, such as CTGF or SERPINE1.

We agree with the referee's comment and have replaced the term “myofibroblasts” by “HOCl-fibroblasts” although in our hands these cells showed a significant increase of α -SMA and type I collagen expression. Also, in order to ensure consistency within the manuscript, we changed the term “myofibroblasts” to “SSc-fibroblasts” in the human co-culture experience (Results section and Figures 4 and 5).

We have performed RT-qPCR on RNA of the co-cultured cells and showed that SSc fibroblasts expressed 6 to 11 time higher amount of SERPINE1: 10.99 ± 0.24 vs 1 ± 0.77 , $p = 0.0002$, 6 times higher amount of CTGF: 6.18 ± 0.28 vs 1 ± 0.57 , $p = 0.001$. Consistent with the TGF- β results, co-culture with LPS^{low} macrophages markedly reduced the expression of those two markers compared to the SSc fibroblasts cultured alone (CTGF: $p = 0.002$; SERPINE1: $p < 0.0001$) while BCG significantly enhanced the production of SERPINE 1 ($p = 0.005$) and tended to increase the expression of CTGF ($p = 0.08$) (Fig. 6f and 6g). The results were confirmed on a second SSc-fibroblasts lineage from a different SSc patient (See supplemental Fig. 7)

Results were added in pages 28-29, lines 631 – 637 as follow:

“Regarding fibrosis markers, mRNA expression of α -SMA, Col I, CD44, ICAM-1, CTGF and SERPINE1 was significantly increased in SSc-fibroblast compared to fibroblast from healthy

controls (HC). Co-culture of PBS-trained macrophages with SSc-fibroblasts did not influence the expression of these genes compared to SSc-myofibroblasts cultured alone. However, when cultured with SSc-fibroblasts, BCG-trained macrophages significantly increased the expression of these markers (except a tendency toward an increase for CTGF (p=0.08)), while LPS^{low}-trained macrophages notably reduced their expression (Fig. 6b-6g).”

The discussion should compare the anticipated signaling of BCG to LPS. The latter is relatively understood and reference(s) regarding the role of TLR4 in SSc pathogenesis should be cited and mentioned briefly. As BCG also activates TLR4 but has other receptors (TLR2), this should also be considered, i.e., why are they seeing different results.

We thank the referee for this suggestion to give more details about the signaling of the two stimuli considered in this paper. In the new version of the manuscript, we mention in the discussion the well-established role of LPS signaling pathway through the TLR4-MD2 complex in the pathogenesis of systemic sclerosis (Page 34, Line 746-748):

“In SSc, it has been clearly established that persistent TLR4-MD2 activation has a pathogenic role in fibrosis progression, with an overexpression on diseased fibroblasts^{20,21}”

Regarding the BCG signaling pathway, it has been shown that the immune training is conferred through a NOD-2 dependent manner. Experiments conducted by Netea et al have shown that the blockade of TLR2 and TLR4 by specific inhibitors did not abolish the training ability induced by BCG while macrophages with a complete NOD-2 deficiency failed to mount an enhanced inflammatory response to BCG training²². Also, the NOD-2 agonist Muramyl dipeptide (MDP) was able to induce immune training. This property was impaired by the inhibition of the RIP2 kinase, a protein of signaling complexes downstream the NOD-2 receptor.

Thus, although TLR2 and TLR4 receptors are involved in the BCG signaling pathway to induce an inflammatory response²³, macrophage immune training require the intracellular NOD-2 receptor that seems to play a central role in this phenomenon. These data are supported by a study demonstrating the role of NOD2 in controlling the growth of Mycobacterium tuberculosis and BCG in human macrophages along with regulating the nature of the inflammatory response²⁴

We added the following paragraph to explain the differences regarding the signaling pathways of our stimulation modes (Page 34, Lines 745-753)

“LPS and BCG signaling share the TLR4 receptor complex which drives the inflammatory responses through MAPK and NF-κB pathways^{69,70}. In SSc, it has been clearly established that persistent TLR4-MD2 activation has a pathogenic role in fibrosis progression, with an overexpression in diseased fibroblasts^{71,72}. LPS tolerance is accompanied by an impairment of the TLR4 signaling with an increase in the negative feedback regulators such as IRAK-M and A20^{15,40}. BCG training has been shown to be dependent of the intracellular NOD2

receptor and the blockade of TLR2 and TLR4 did not abolish the training ability induced by BCG¹⁰. Thus it appears that our stimuli model uses a different signaling pathway that could contribute to explain the opposite effects obtained”.

References:

1. Arts, R. J. W. *et al.* Immunometabolic Pathways in BCG-Induced Trained Immunity. *Cell Rep.* **17**, 2562–2571 (2016).
2. Cheng, S.-C. *et al.* Broad defects in the energy metabolism of leukocytes underlie immunoparalysis in sepsis. *Nat. Immunol.* **17**, 406–413 (2016).
3. Arts, R. J. W., Joosten, L. A. B. & Netea, M. G. Immunometabolic circuits in trained immunity. *Semin. Immunol.* **28**, 425–430 (2016).
4. Novakovic, B. *et al.* β -Glucan Reverses the Epigenetic State of LPS-Induced Immunological Tolerance. *Cell* **167**, 1354-1368.e14 (2016).
5. Saeed, S. *et al.* Epigenetic programming of monocyte-to-macrophage differentiation and trained innate immunity. *Science* **345**, 1251086 (2014).
6. Seeley, J. J. & Ghosh, S. Molecular mechanisms of innate memory and tolerance to LPS. *J. Leukoc. Biol.* **101**, 107–119 (2017).
7. Josefsen, K., Nielsen, H., Lorentzen, S., Damsbo, P. & Buschard, K. Circulating monocytes are activated in newly diagnosed type 1 diabetes mellitus patients. *Clin. Exp. Immunol.* **98**, 489–493 (1994).
8. Gjelstrup, M. C. *et al.* Subsets of activated monocytes and markers of inflammation in incipient and progressed multiple sclerosis. *Immunol. Cell Biol.* **96**, 160–174 (2018).
9. Li, Y., Lee, P. Y. & Reeves, W. H. Monocyte and Macrophage Abnormalities in Systemic Lupus Erythematosus. *Arch. Immunol. Ther. Exp. (Warsz.)* **58**, 355–364 (2010).
10. Vogel, D. Y. *et al.* Macrophages in inflammatory multiple sclerosis lesions have an intermediate activation status. *J. Neuroinflammation* **10**, 35 (2013).

11. Alkanani, A. K. *et al.* Dysregulated Toll-like receptor-induced interleukin-1 β and interleukin-6 responses in subjects at risk for the development of type 1 diabetes. *Diabetes* **61**, 2525–2533 (2012).
12. Steinbach, F. *et al.* Monocytes from systemic lupus erythematosus patients are severely altered in phenotype and lineage flexibility. *Ann. Rheum. Dis.* **59**, 283–288 (2000).
13. Lioté, F., Boval-Boizard, B., Weill, D., Kuntz, D. & Wautier, J. L. Blood monocyte activation in rheumatoid arthritis: increased monocyte adhesiveness, integrin expression, and cytokine release. *Clin. Exp. Immunol.* **106**, 13–19 (1996).
14. Varga, J. & Abraham, D. Systemic sclerosis: a prototypic multisystem fibrotic disorder. *J. Clin. Invest.* **117**, 557–567 (2007).
15. Servettaz, A. *et al.* Selective oxidation of DNA topoisomerase 1 induces systemic sclerosis in the mouse. *J. Immunol. Baltim. Md 1950* **182**, 5855–5864 (2009).
16. Vona, R. *et al.* Oxidative stress in the pathogenesis of systemic scleroderma: An overview. *J. Cell. Mol. Med.* **22**, 3308–3314 (2018).
17. Abdulle, A. E., Diercks, G. F. H., Feelisch, M., Mulder, D. J. & Goor, H. van. The Role of Oxidative Stress in the Development of Systemic Sclerosis Related Vasculopathy. *Front. Physiol.* **9**, (2018).
18. Allanore, Y., Borderie, D., Lemaréchal, H., Ekindjian, O. G. & Kahan, A. Nifedipine decreases sVCAM-1 concentrations and oxidative stress in systemic sclerosis but does not affect the concentrations of vascular endothelial growth factor or its soluble receptor 1. *Arthritis Res. Ther.* **6**, R309–R314 (2004).
19. Witko-Sarsat, V. *et al.* Advanced oxidation protein products as a novel marker of oxidative stress in uremia. *Kidney Int.* **49**, 1304–1313 (1996).
20. Bhattacharyya, S. *et al.* TLR4-dependent fibroblast activation drives persistent organ fibrosis in skin and lung. *JCI Insight* **3**, (2018).
21. Stifano, G. *et al.* Chronic Toll-like receptor 4 stimulation in skin induces inflammation, macrophage activation, transforming growth factor beta signature gene expression, and fibrosis. *Arthritis Res. Ther.* **16**, R136 (2014).

22. Kleinnijenhuis, J. *et al.* Bacille Calmette-Guerin induces NOD2-dependent nonspecific protection from reinfection via epigenetic reprogramming of monocytes. *Proc. Natl. Acad. Sci. U. S. A.* **109**, 17537–17542 (2012).
23. Basu, J., Shin, D.-M. & Jo, E.-K. Mycobacterial signaling through toll-like receptors. *Front. Cell. Infect. Microbiol.* **2**, (2012).
24. Landes, M. B., Rajaram, M. V. S., Nguyen, H. & Schlesinger, L. S. Role for NOD2 in Mycobacterium tuberculosis-induced iNOS expression and NO production in human macrophages. *J. Leukoc. Biol.* **97**, 1111–1119 (2015).

REVIEWERS' COMMENTS:

Reviewer #1 (Remarks to the Author):

The authors have responded appropriately to my comments.

Reviewer #2 (Remarks to the Author): HAD NO COMMENTS TO AUTHORS

Response to referees

REVIEWERS' COMMENTS:

Reviewer #1 (Remarks to the Author):

The authors have responded appropriately to my comments.

Reviewer #2 (Remarks to the Author): HAD NO COMMENTS TO AUTHORS

The authors thank the reviewers for their time in reviewing the manuscript. The recommendations, suggestions and fruitful exchanges greatly contributed to improve it significantly.